# TECHNIQUES AND RESOURCES

# Developmental dynamics of catshark cranial neural crest cells provide insights into gnathostome facial evolution

Elio Escamilla-Vega[1], Andrea P. Murillo-Rincón[1], Louk W. G. Seton[1], Ann-Katrin Koch[2,3], Stella Kyomen[1], Carsten Fortmann-Grote[1], Jörg U. Hammel[4], Timo Moritz[2,5] and Markéta Kaucká[1,*]

## ABSTRACT

Cranial neural crest cells (CNCCs) are a vertebrate-specific, multipotent cell population central to facial morphogenesis and a cellular substrate for evolutionary change. Although core CNCC developmental programmes are deeply conserved, changes in their gene expression programmes and cell behaviour underlie both macroevolutionary transitions and microevolutionary adaptations. While CNCC biology has been well characterized in bony vertebrates, comparatively little is known about CNCC properties and the behaviour of their derivatives in cartilaginous fishes (Chondrichthyes). To address this gap, we investigate CNCC development in a representative chondrichthyan: the small-spotted catshark (*Scyliorhinus canicula*). By integrating high-resolution molecular and morphological analyses, we reveal how conserved developmental programmes are modulated in chondrichthyans to generate divergent facial morphologies. We show that the molecular toolkit of CNCC is largely conserved across jawed vertebrates, and the developmental divergence and lineage-specific differences arise from divergent behaviour of their ectomesenchymal derivatives. These findings establish a high-resolution reference of CNCC biology in Chondrichthyes and uncover the evolutionary origins of both shared and lineage-specific traits, offering key insights into the developmental and evolutionary processes shaping gnathostome facial diversity.

KEY WORDS: Chondrichthyes, Small-spotted catshark, *Scyliorhinus canicula*, Cranial neural crest cells, Facial morphogenesis, Single-cell transcriptomics

## INTRODUCTION

Neural crest cells (NCCs) are a vertebrate-specific transient population of multipotent cells that emerge during embryogenesis and give rise to numerous cell types and tissues. The emergence of NCCs fuelled the evolution of early vertebrates and enabled major evolutionary innovations such as the acquisition of jaws, craniofacial skeleton and improved sensory capabilities (Martik and Bronner, 2021). Beyond these macroevolutionary innovations, NCCs contribute to the development of many highly evolvable lineage-specific traits, such as the beak in birds (Schneider and Helms, 2003), pigmentation in cichlid fishes (Marconi et al., 2024), and, most notably, the jaws and facial skeleton in jawed vertebrates (gnathostomes) (Santagati and Rijli, 2003). Although NCCs are highly conserved in terms of developmental origin, differentiation potential and core gene expression, particularly in gnathostomes, the modulation of where and when these cells emerge, and how their derivatives behave during development significantly contributes to vertebrate facial diversity (Martik and Bronner, 2021; Martik et al., 2019; Monroy et al., 2022; Stundl et al., 2020; Wakamatsu et al., 2014). Such alterations arise from changes in conserved gene regulatory networks (GRNs) that modulate NCC development or properties of their derivatives, thus generating morphological variation.

Based on the axial position of their origin, NCCs are classified into four main subpopulations that exhibit distinct developmental potential: cranial, vagal, trunk and sacral (Rothstein et al., 2018). Notably, cranial NCCs (CNCCs) are the only subpopulation with skeletogenic potential (Cordero et al., 2011). As such, CNCCs contribute to the facial skeleton, which ultimately dictates the shape of the face. CNCCs follow three main migratory streams – trigeminal (also known as mandibular), hyoid and branchial (Minoux and Rijli, 2010; Santagati and Rijli, 2003; Square et al., 2017) – that are highly conserved across vertebrates.

While the expression of core genes and signalling events implicated in CNCC development is highly conserved across vertebrates, notable spatiotemporal variation in gene expression is observed across species, which has been linked to the formation of species-specific traits (Albertson and Kocher, 2006; Monroy et al., 2022; Stundl et al., 2020; Wakamatsu et al., 2014). After migration, CNCCs differentiate into distinct cellular lineages, including neurons and glia of the peripheral nervous system (PNS), pigment cells and ectomesenchymal tissues (Martik and Bronner, 2021; Rothstein et al., 2018). Ectomesenchymal cells give rise to cartilage and bone, consequently creating the embryonic blueprint of the future craniofacial skeleton. Experimental modulation of ectomesenchyme development, particularly cell behaviour such as polarity or proliferation, leads to altered facial geometries while maintaining the general organization of the skeletal elements (Kaucka et al., 2017). In contrast, perturbations at the level of CNCC specification, delamination and migration often result in broad phenotypes, simultaneously affecting multiple derivatives. Thus, while modifications of CNCC GRNs influence a wide range of tissues, evolutionary changes within the ectomesenchymal lineage result in the specific modulation of the craniofacial skeleton and adaptation to ecological niches (Abzhanov et al., 2004; Usui and Tokita, 2018). Collectively, these findings support the view that the cellular behaviour, patterning and differentiation potential of the

[1]Max Planck Institute for Evolutionary Biology, August-Thienemann-Str. 2, 24306 Plön, Germany. [2]Ocean Museum Germany, Katharinenberg 14–20, 18439 Stralsund, Germany. [3]Institute of Biosciences, University of Rostock, Albert-Einstein-Str. 3, 18059 Rostock, Germany. [4]Institute of Materials Physics, Helmholtz-Zentrum Hereon, Max-Planck-Str. 1, 21502 Geesthacht, Germany. [5]Leibniz Institute for the Analysis of Biodiversity Change, Martin-Luther-King-Platz 3, D-20146 Hamburg, Germany.

*Author for correspondence (kaucka@evolbio.mpg.de)

E.E., 0009-0005-6106-0802; A.P.M., 0009-0003-7532-3736; L.W.G.S., 0009-0000-6946-7686; A.-K.K., 0000-0001-8810-9171; S.K., 0000-0003-0105-4636; C.F., 0000-0002-2579-5546; J.U.H., 0000-0002-6744-6811; T.M., 0000-0003-1281-7432; M.K., 0000-0002-8781-9769

ectomesenchyme are targets for species-specific craniofacial adaptations (Fabian and Crump, 2023; Kyomen et al., 2023).

Comparative studies across gnathostomes have provided us with a fundamental understanding of how changes in CNCCs fuel diversification and speciation. Model organisms like mouse (Soldatov et al., 2019), chicken (Le Douarin, 2004), zebrafish (Fabian et al., 2022) and clawed frogs (Medina-Cuadra and Monsoro-Burq, 2021) dominate comparative and developmental research but, recently, non-model species like squamate reptiles (Pranter and Feiner, 2025), marsupials (Wakamatsu et al., 2014) and non-teleost ray-finned fishes (Stundl et al., 2020) have provided a broader evolutionary perspective. However, CNCC biology studies have primarily focused on bony vertebrates (Osteichthyes), leaving a pivotal phylogenetic node, Chondrichthyes (cartilaginous fishes), largely unexplored, thus hindering the investigation of CNCC evolutionary history at the onset of gnathostomes.

Here, we set out to characterize the molecular signature and developmental dynamics of CNCCs and their ectomesenchymal derivatives in the small-spotted catshark (catshark, *Scyliorhinus canicula*), uncovering the spatiotemporal aspects of transcriptional programmes and cell behaviour contributing to the acquisition of the chondrichthyan facial morphology. We combined single-cell transcriptomics with whole-mount laser confocal microscopy and synchrotron radiation micro-computed tomography (SRμCT) along catshark ontogeny. By detailed spatiotemporal and molecular characterization of CNCCs in the catshark, we reveal the conserved transcriptional profile of CNCCs, and their distinctive migratory routes, unique facial ectomesenchyme behaviour and progressive differentiation into the cartilaginous elements that form the shark facial skeleton. Additionally, we present an information-rich reference, together with an open-access, interactive database that will serve as a valuable resource for comparative studies, providing essential insights into the early evolution of CNCCs and their role in vertebrate craniofacial diversity.

## RESULTS
### CNCC migratory streams are conserved in the catshark
To understand the spatiotemporal dynamics of CNCCs in the catshark, we mapped CNCCs and their migratory streams using SOX9 immunofluorescence. Specifically, we focused on consecutive developmental stages from the onset of neurulation, when CNCCs are specified, to the establishment of pharyngeal arches and basic embryonic head morphology (stages, St.14-25) (Fig. 1).

In the catshark, neurulation begins at St.14, when SOX9 expression first appears in the cephalic region (Fig. 1A), labelling cells at the neural plate border in the developing head, as well as in the trunk notochord (Fig. 1B-B'). By St.15, SOX9 labels specified CNCC in the elevating neural folds, marking the origin of the trigeminal stream (Fig. 1C,G), while the neural plate remains open along the anteroposterior axis (Fig. 1A,G). Only notochord cells are labelled in the trunk (Fig. 1C'). At St.16, the elevated neural folds in the trunk region fuse, while the anterior and posterior neuropores remain open. The trigeminal and hyoid CNCCs are specified within the dorsal region of the anterior neuropore but have not yet delaminated (Fig. 1A,D,D',G). At late St.16, the otic placode becomes visible in the hyoid region (Fig. 1D",G). Following neural tube closure at St.17, trigeminal CNCC delaminate and initiate migration (Fig. 1A,E-E",G). By early St.18, the trigeminal stream divides into mandibular and frontonasal branches (Fig. 1A,G). Mandibular CNCCs migrate exclusively dorsolaterally to populate the first pharyngeal arch, while frontonasal CNCCs migrate along the midline toward the upper periocular region (Fig. 1A,G). A subset of frontonasal CNCCs

migrate dorsolaterally alongside mandibular CNCCs before turning rostrally towards the lower periocular area (Fig. 1A,G). Hyoid CNCCs are slightly more advanced compared to the branchial stream. By early St.18, hyoid CNCCs begin delamination, and migrate by mid-St.18 (Fig. 1A,F'-G). Meanwhile branchial CNCCs begin delaminating at mid-St.18 and migrate towards their respective pharyngeal arches by late St.18 (Fig. 1A,G). The last migrating trigeminal CNCCs are observed at St.19, and remain in a periocular position at St.20 before the SOX9 expression declines (Fig. 1G,H). The last migrating hyoid CNCCs are observed at St.20 (Fig. 1A,G).

At later embryonic stages (St.21-25), branchial CNCCs continue migrating to populate the pharyngeal arches (Fig. 1H). The pharyngeal arches form sequentially cranio-caudally with the first arch becoming visible at St.18, and the seventh arch by St.25. Each pharyngeal arch consists of a mesodermal core surrounded by CNCCs, which give rise to the pharyngeal ectomesenchyme (Fig. 1I-K'). CNCCs retain high SOX9 expression as they colonize the newly formed arches (e.g. seventh pharyngeal arch is SOX9$^+$ at St.25) (Fig. 1H-K'). However, this expression becomes restricted over time, with SOX9 later persisting only in specific ectomesenchymal subpopulations (Fig. 1K').

In summary, CNCC delamination in the catshark occurs upon closure of the neural tube, and their migratory streams closely resemble those described in other vertebrate species, comprising three major streams: trigeminal, hyoid and branchial. In the catshark, the trigeminal stream is the largest stream, and is further subdivided into distinct mandibular and frontonasal branches. The most notable difference lies in the migration and final positioning of the frontonasal branch (also known as premandibular) (Debiais-Thibaud et al., 2013), which remains localized in the periocular area rather than rapidly migrating to the anteriormost region of the face, as observed in Osteichthyes (Minoux and Rijli, 2010; Theveneau and Mayor, 2012).

### Single-cell transcriptomic analysis of the catshark embryonic head
While NCC single-cell transcriptomics in conventional osteichthyan species have often been combined with fluorescence-activated cell sorting of labelled NCCs either upon electroporation or using transgenic reporter lines (Fabian et al., 2022; Soldatov et al., 2019; Tatarakis et al., 2021; Williams et al., 2022), these approaches are not currently feasible in Chondrichthyes due to technological and biological bottlenecks. Hence, to characterize the molecular signature and early cell fate specifications of CNCCs in the catshark that contribute to facial morphogenesis, we performed single-cell RNA sequencing (scRNA-seq) of the developing head (Fig. 2A). We microdissected the catshark embryonic head below the otic capsule; therefore, the resulting dataset comprised primarily the trigeminal CNCCs. Given our focus on facial development and evolution, we intentionally restricted the single-cell analyses to trigeminal CNCCs. Hyoid and branchial CNCCs, which primarily contribute to the pharyngeal (gill) arches, were therefore not included (Martik and Bronner, 2021; Santagati and Rijli, 2003; Sleight and Gillis, 2020). We profiled five key developmental stages: (1) St.16, specified pre-migratory CNCCs; (2) St.17, delaminating CNCCs; (3) St.18, migrating CNCCs; (4) St.19, last migrating trigeminal CNCCs; and (5) St.20, early differentiating CNCCs. Due to the limited availability of catshark embryos, these five developmental stages were pooled together to acquire a sufficient number of cells.

Upon stringent quality control (Fig. S1; see Materials and Methods), a total of 6585 single-cell transcriptomes were recovered. Cell clustering and dimensionality reduction were applied and the

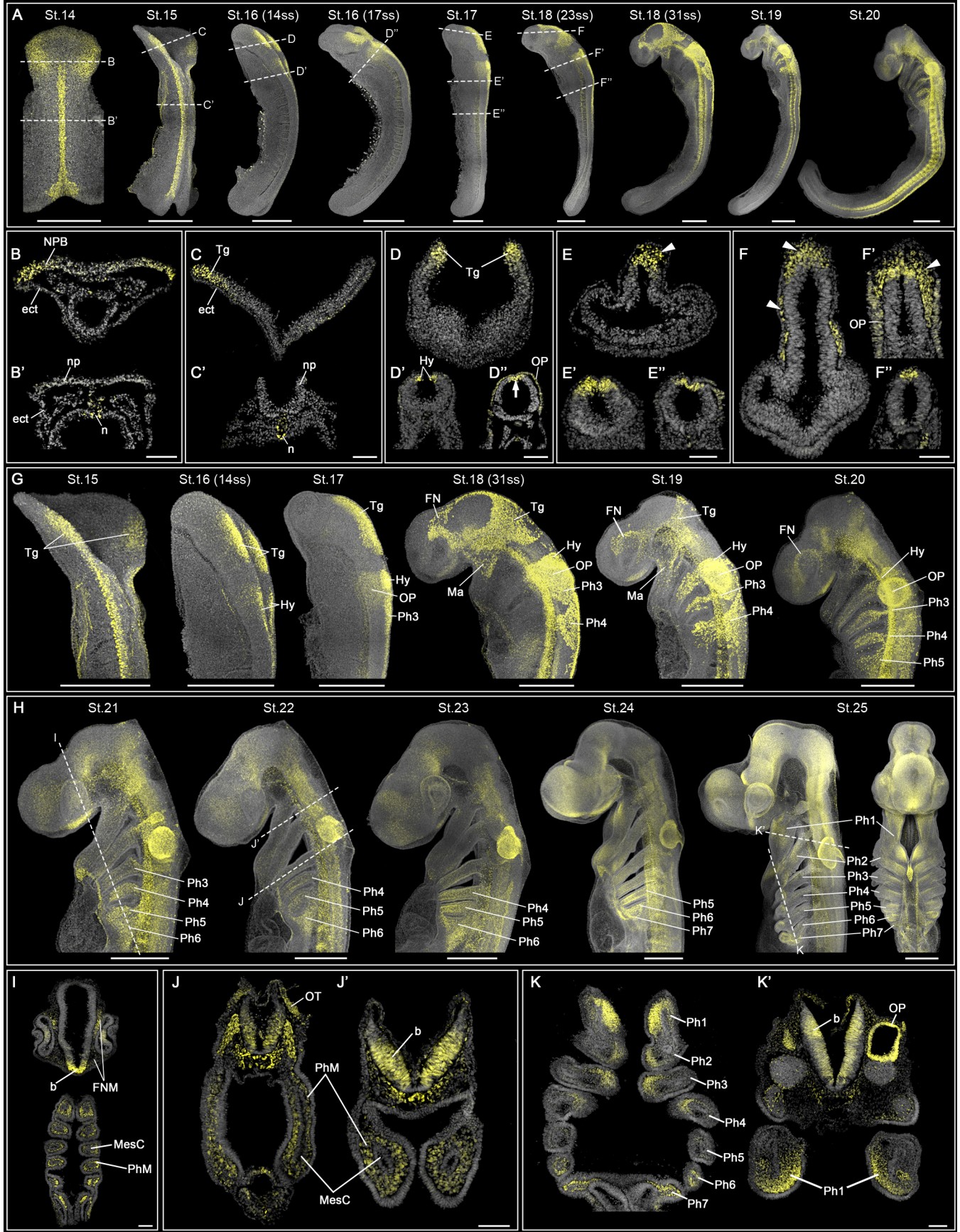

**Fig. 1.** See next page for legend.

**Fig. 1. CNCC migratory patterns in the catshark revealed by SOX9 immunofluorescence.** (A-F″) St.14-20 catshark embryos (somite numbers in brackets). White dashed lines indicate the position of sections shown in B-F″. (E-F″) White arrowheads indicate delaminating (E) and migratory (F,F′) CNCCs. White arrow in D″ indicates the fused neural tube at the trunk region. (G) Higher magnifications of the head region from embryos shown in A. (H-K′) Catshark during pharyngeal arch formation (St.21-25). White dashed lines in H indicate the position of sections shown in I-K′. Note the ectomesenchymal spatial domains where SOX9 remains expressed in the Ph1 at St.25 (K′). Scale bars: 500 μm (wholemount) and 100 μm (sections). b, brain; ect, ectoderm; FN, frontonasal branch; FNM, frontonasal ectomesenchyme; Hy, hyoid stream; Ma, mandibular branch; MesC, mesodermal core; n, notochord; np, neural plate; NPB, neural plate border; OP, otic placode; Ph1-7, pharyngeal arches 1-7; PhM, pharyngeal arch ectomesenchyme; ss, somites; Tg, trigeminal stream.

data were visualized by Uniform Manifold Approximation and Projection (UMAP) (Fig. 2B). The recovered 15 major cell populations were annotated using canonical marker genes and differential gene expression analysis (Fig. 2C). The 15 main cell clusters correspond to all major anatomical structures and cell types expected in the early developing head, including CNCCs, central nervous system (CNS), blood vessels, endoderm, mesoderm and ectoderm. To validate cluster identities, we performed whole-mount hybridization chain reaction (HCR) *in situ* hybridization using St.19 embryos as a representative stage for the dataset (Fig. 2D). The CNCC cluster is composed of $HOX^-$ cells (Fig. S2), corresponding to the trigeminal CNCC subpopulation (Oulion et al., 2011; Soldatov et al., 2019).

## Catshark CNCCs exhibit conserved developmental dynamics

To investigate the transcriptional programmes of catshark CNCC in greater detail, we subset and analysed the CNCC cluster independently (Fig. 3A). The increased clustering resolution allowed the identification of five distinct subclusters. Three of these subclusters correspond to bona fide CNCCs of the prospective mesenchymal lineage (ectomesenchymal progenitors and ectomesenchymal cells), while the remaining two represent unrelated populations: the hatching gland and $GAD1^+$ neurons (Fig. 3B).

The hatching gland is a specialized ectodermal structure that facilitates the hatching process. We used the canonical marker *XBP1* to validate the identity of this subcluster and confirmed its expression in the prospective hatching gland (Fig. S4). In *Xenopus* sp., both the hatching gland and CNCCs arise from the neural plate border, which may explain their co-clustering at lower resolution (Fig. 2B). The *GAD1* gene is essential for GABAergic signalling, and its loss of function leads to craniofacial defects (O'Connor et al., 2018; Oh et al., 2010), which has raised the hypothesis about its role in CNCC biology (O'Connor et al., 2018). However, in the catshark, similar to other gnathostomes, *GAD1* expression was mapped to brain regions but not to CNCCs (Fig. S5) (Lüffe et al., 2021; Maddox and Condie, 2001). These findings suggest that while *GAD1* is important for craniofacial development and is conserved across gnathostomes, its effect on CNCCs is likely indirect (Oh et al., 2010).

The remaining three subclusters represent bona fide CNCC subpopulations. One subcluster comprises specified, delaminating and migratory, yet undifferentiated, cells, collectively termed 'early CNCC'. These cells express canonical CNCC markers (*SOX9*, *SOX10*, *FOXD3* and *EDNRB*) but lack markers of fate commitment (*MITF*, *PHOX2B*, *NEUROG1*, *NEUROG2* and *TWIST1*) (Fig. 3C). The other two subclusters, ectomesenchymal progenitors and ectomesenchymal cells, represent progressive stages of the ectomesenchymal lineage. Ectomesenchymal cells

express high levels of mesenchyme-specifying genes (*TWIST1*, *TWIST2*, *FLI1* and *EDNRA*). In contrast, ectomesenchymal progenitors are characterized by reduced expression of both NCCs and mesenchymal markers, suggesting a transitional state (Fig. 3C). To reconstruct and estimate the temporal dynamics of the mesenchymal lineage, we performed a pseudotime analysis (Fig. 3D), which revealed that ectomesenchymal cells have higher pseudotime values, while ectomesenchymal progenitors and early CNCCs display lower pseudotime values. The inferred pseudotime trajectory reflects the gradual commitment of CNCCs to mesenchymal fate, marked by the progressive upregulation of key transcription factors and signalling receptors associated with ectomesenchyme differentiation, alongside downregulation of early NCC genes (Fig. 3E). Given the transcriptional similarities to the murine and zebrafish CNCC developmental programmes, collectively, these results support the evolutionary conservation of the CNCC-ectomesenchyme differentiation trajectory across gnathostomes (Soldatov et al., 2019; Tatarakis et al., 2021).

Since CNCCs are influenced by both intrinsic gene expression and extrinsic cues (Tatarakis et al., 2021), we examined potential signalling interactions between CNCC and their surrounding tissues using CellChat, a computational tool for inferring intercellular communications from scRNA-seq data (Jin et al., 2021) (see Materials and Methods; Fig. S6). We focused on signalling pathways relevant to CNCC subpopulations (Fig. 3F), particularly on signalling mediated by secreted ligands. CNCCs primarily acted as signal receivers, while ectodermal and CNS populations served as the main signal senders (Fig. 3F). Key ectodermal signals included BMPs, PDGF and SEMA3 pathways, while CNS-derived signals included WNT, SLIT and EDN, all known to guide CNCC migration and behaviour in Osteichthyes (Jia et al., 2005; Smith and Tallquist, 2010; Square et al., 2020; Theveneau and Mayor, 2014).

We also identified putative novel signalling sources and pathways previously not reported in other species, including periostin (POSTN) secreted by the notochord (Fig. 3F). To determine whether notochord-derived *POSTN* represents a conserved gnathostome feature or a lineage-specific adaptation in Chondrichthyes, we screened available single-cell transcriptomic datasets from representative gnathostomes (mouse, chicken, Western clawed frog and zebrafish) to identify the cellular sources of *POSTN* during embryogenesis (Table S1). In chicken and the Western clawed frog, *POSTN* shows strong expression in the notochord, whereas in mouse and zebrafish it is primarily restricted to mesodermal tissues, epithelial cells and the gut (Table S1). To validate these findings, we examined *POSTN* expression in catshark, mouse and chicken embryos (Fig. 4). In catshark, *POSTN* is strongly expressed in the notochord, with weaker expression the endoderm and heart (Fig. 4A). Similarly, high *POSTN* expression is detected in the chicken notochord (Fig. 4B), whereas no comparable signal is observed in mouse (Fig. 4C), consistent with the single-cell data (Table S1). Together, these results suggest lineage-specific differences in the deployment of the POSTN signalling pathway, particularly in mammals and actinopterygians relative to other gnathostome lineages. However, functional validations will be required in the future to determine its contribution to lineage-specific facial morphologies.

Taken together, our analyses of CNCC developmental trajectories reveal a striking evolutionary conservation across gnathostomes, indicating a deeply conserved molecular toolkit. Nonetheless, differences in CNCC interactions with their surrounding tissues are also observed, which may be related to lineage-specific adaptations (e.g. notochord-secreted *POSTN*). Given the remarkable level of conservation in the CNCC molecular programme, we hypothesized

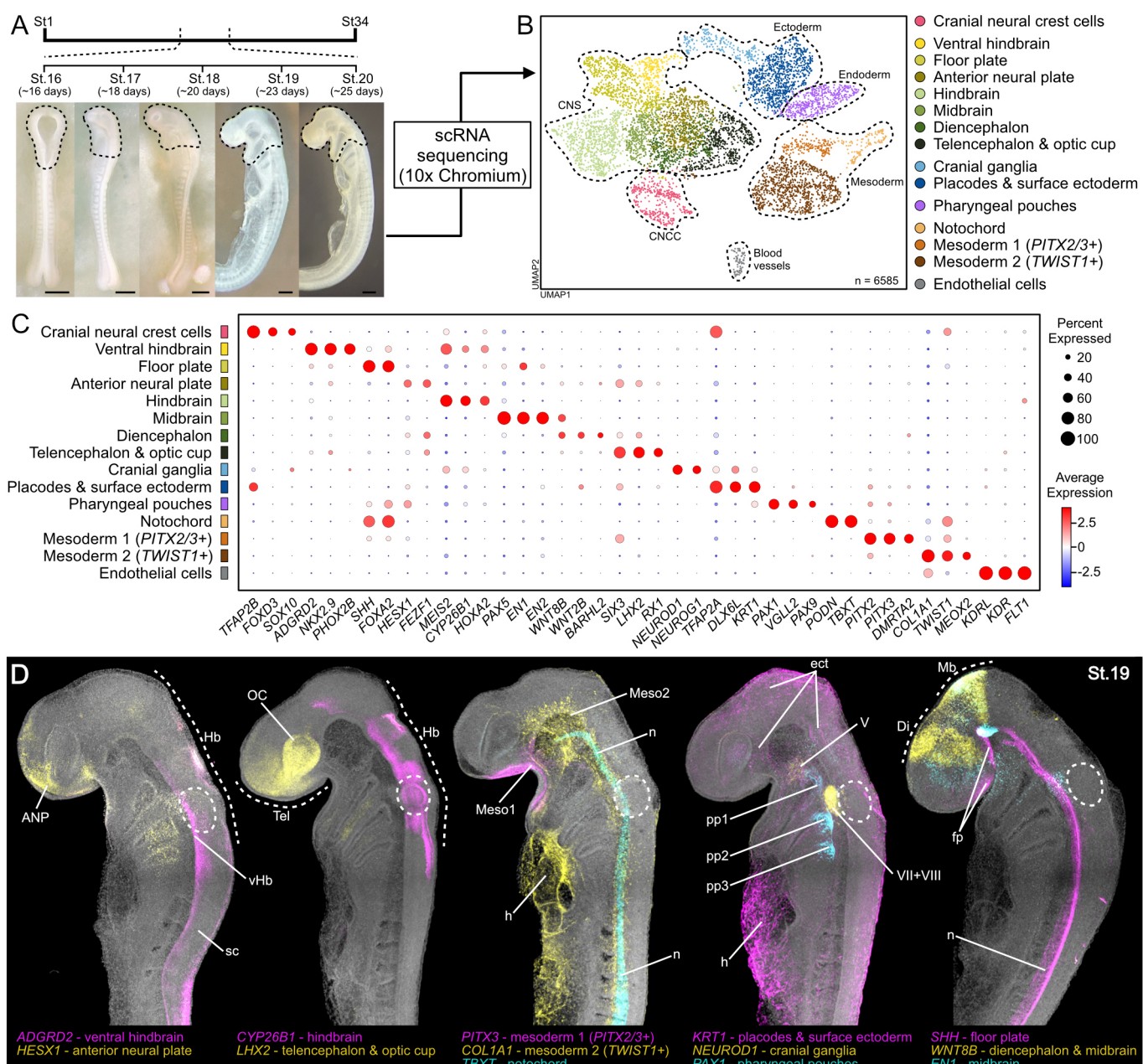

**Fig. 2. Catshark developing head at single-cell resolution.** (A) Representative catshark embryos from stages selected for scRNA-seq. Black dashed lines outline the areas microdissected for the single-cell experiment. Scale bars: 100 µm. (B) UMAP visualization of the single-cell dataset. (C) Dotplot showing main marker genes used for cluster annotation in B. (D) *In situ* HCR for selected marker genes in St.19 catshark embryos (B,C). White dashed circles indicate the otic capsule. For endothelial cells, *FLI1* and *KDR* expression was detected in the developing blood vessels (Fig. S3). Scale bar: 500 µm. ANP, anterior neural plate; Di, diencephalon; ect, ectoderm; fp, floor plate; h, heart; Hb, hindbrain; Mb, midbrain; Meso1, mesoderm 1; Meso2, mesoderm 2; n, notochord; OC, optic cup; pp1-3, first to third pharyngeal pouches; sc, spinal cord; Tel, telencephalon; V, trigeminal ganglia; vHb, ventral hindbrain; VII+VIII, acousticofacial ganglia.

that developmental divergence may not arise within CNCCs themselves but rather in their subsequent derivatives or surrounding tissues.

## CNCC-derived neuroglia are not detected in the scRNA-seq analysis

CNCCs also give rise to neuroglia derivatives, including sensory and autonomic neurons and glial cells of the PNS (Cordero et al., 2011; Martik and Bronner, 2021). To investigate this cell lineage, we first examined the expression of *PHOX2B*, a canonical marker of autonomic neurons (Fig. S7A). Consistent with our initial cluster annotation, *PHOX2B*$^+$ cells were detected in the ventral hindbrain (Fig. 2C and Fig. S7A). Notably, *PHOX2B* expression in the cranial ganglia, presumably marking autonomic neurons, was only observed from St.22 onwards (Fig. S7A), beyond the developmental window covered in the scRNA-seq experiment (Fig. 2A). Next, to assess the presence of CNCC-derived neuroglia in the scRNA-seq dataset, potentially clustered separately from the main CNCC populations, we screened for cells expressing neuroglial markers. We queried the full scRNA-seq dataset using a panel of established markers (Fig. S7B,C) (Soldatov et al., 2019). No molecular signatures for autonomic

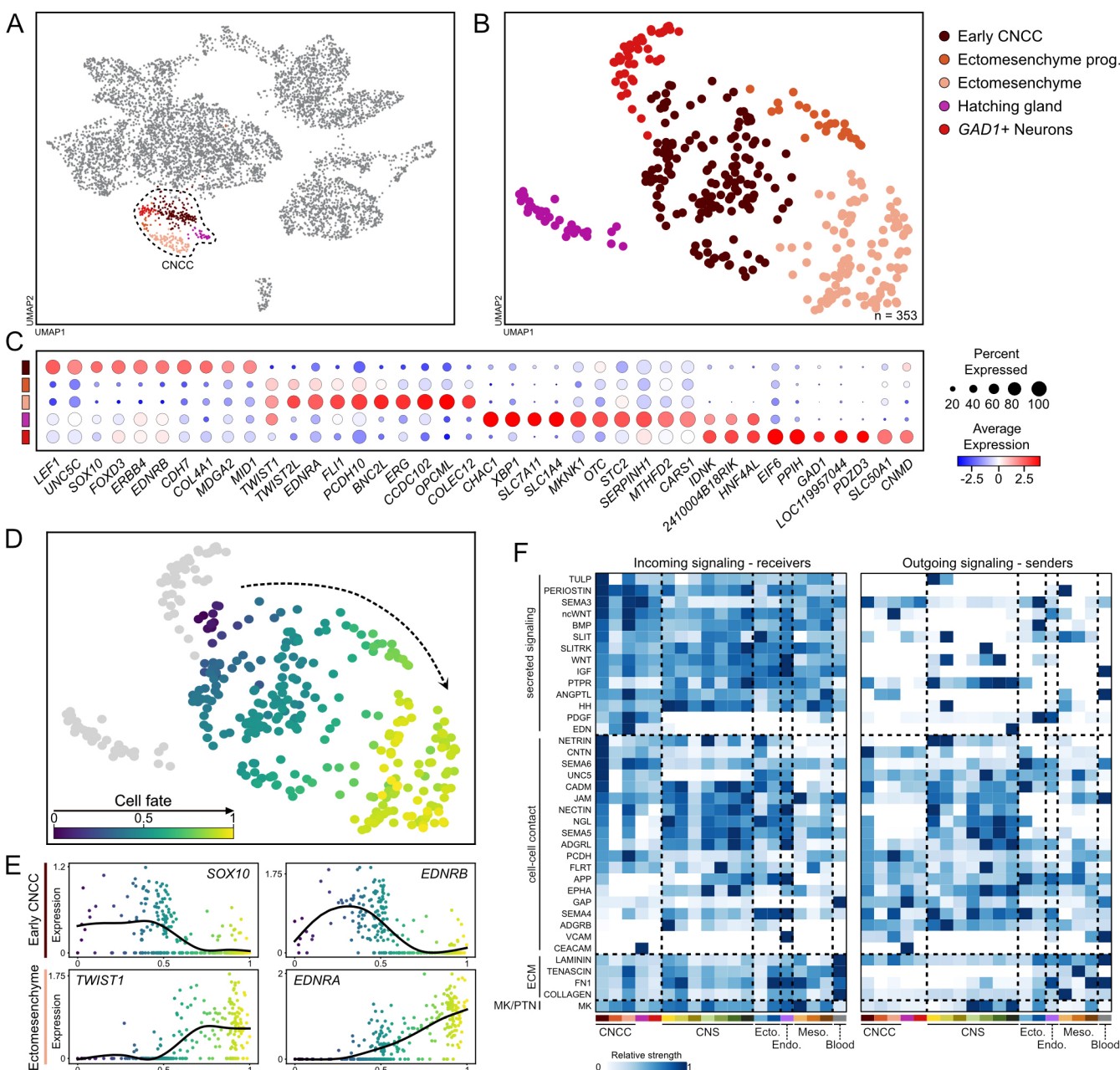

**Fig. 3. CNCC subset from the whole-head scRNA-seq dataset.** (A,B) UMAP of the entire scRNA-seq (A) and higher resolution subset (B). (C) Dotplot showing main marker genes used for cell cluster annotation in B. (D) Pseudotime analysis of inferred developmental progression, purple to yellow represent from a less to a more specified cell fate, respectively. (E) Pseudotime expression trends of representative genes from C. (F) CellChat cell-cell communication predictions. Incoming signalling refers to the signals received by the target populations (receivers) that are sent by sender populations (outgoing signalling). Only statistically significant signalling interactions are shown ($P<0.05$, Wilcoxon rank sum and one-sided permutation tests). ECM, extracellular matrix; ecto, ectoderm; endo, endoderm; meso, mesoderm; MK/PTN, midkine and pleiotrophin; prog, progenitors.

neurons or glial cells were detected in the dataset. However, we detected a small cell population expressing early sensory neuronal markers (*ISL1*, *NEUROG1*, *INSM1* and *POU4F1*) within the cranial ganglia, likely reflecting the presence of placode-derived sensory neurons (Fig. S7C). In mice and zebrafish, CNCCs commit sequentially first into ectomesenchyme (early-migratory) and then into neuroglia (late-migratory) (Erickson et al., 2024 preprint; Tatarakis et al., 2021). Such late-migratory neuroglia-committed CNCCs are likely present at late St.20 in the catshark, and might not been fully captured in the presented dataset. Additionally, this cell lineage may emerge at later developmental stages (St.21-22) that were

not present in the scRNA-seq dataset. Hence, the apparent absence of CNCC-derived neuroglia in the scRNA-seq dataset may be due to developmental timing, and further research is required to pinpoint the exact developmental stage when CNCC-derived neuroglia emerge in the catshark.

## The periocular ectomesenchyme of the catshark is homologous to the frontonasal ectomesenchyme in Osteichthyes

During early craniofacial development, ectomesenchymal cells proliferate and create the frontonasal ectomesenchyme, which, in

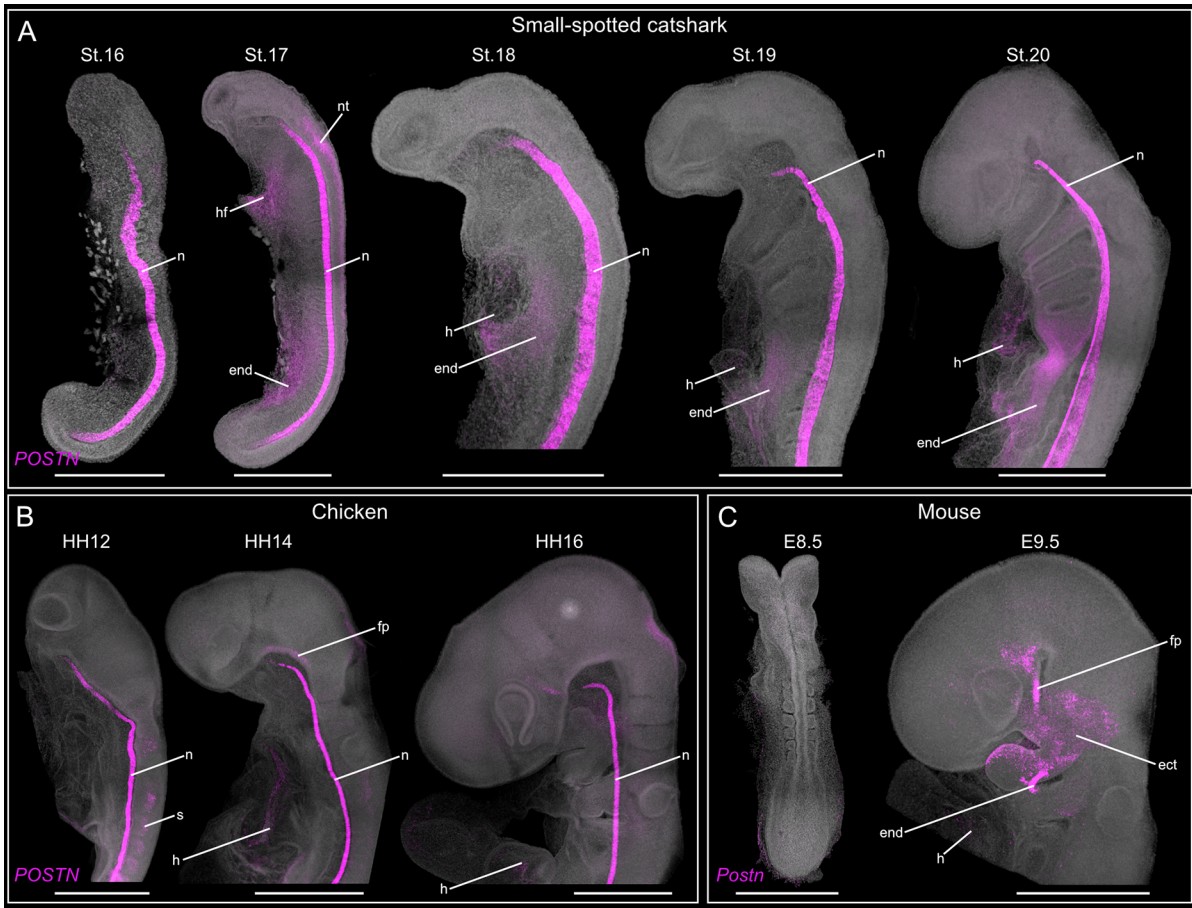

**Fig. 4. *POSTN* expression pattern during embryogenesis in representative gnathostome species.** (A-C) *POSTN* expression during catshark (A), chicken (B) and mouse (C) development. Note that *Postn* expression is not observed in the mouse notochord. Scale bars: 500 μm. ect, ectoderm; end, endoderm; fp, floor plate; h, heart; hf, heart field; n, notochord; nt, neural tube; s, somites.

amniotes, gives rise to distinct facial prominences that fuse to shape the face (Abramyan and Richman, 2015). The spatial and temporal dynamics of ectomesenchyme growth critically influence facial morphology and contribute to species-specific craniofacial structures (Murillo-Rincón and Kaucka, 2020; Usui and Tokita, 2018). To investigate whether ectomesenchyme dynamics in the catshark could be a source of developmental divergence, we examined the expression patterns of mesenchymal marker genes across embryogenesis to map the facial ectomesenchyme and elucidate the catshark-specific facial patterning (Fig. 5).

We first mapped the expression of the mesenchymal-fate specifying gene *TWIST1*, alongside two other traditional marker genes *PAX7* and *WNT1* (Fig. 5A). Consistent with the findings in other vertebrate species (Bedois et al., 2024; Murdoch et al., 2012), *PAX7* expression was detected in the CNS and migratory CNCCs at St18-20, while *WNT1* was primarily restricted to the hindbrain-midbrain boundary (Fig. S8). *TWIST1* was absent prior to CNCC migration (St.16-17), but became detectable at St.18 in mesodermal tissues and small cell populations within the first pharyngeal arch and periocular region (Fig. 5A and Fig. S8). These small *TWIST1*⁺ regions align with sites populated by newly arrived CNCCs at St.18 (Fig. 1G), indicating the first signs of ectomesenchymal fate-commitment in these locations. By St19-20, CNCC migration was largely complete, with strong *TWIST1* expression in the pharyngeal arches and the periocular region (Fig. 5A). At later stages, *TWIST1* expression in the periocular region declined and became spatially restricted within the pharyngeal arches. This transition suggests that CNCCs activate mesenchymal programmes upon reaching their target sites, with the periocular domain functioning as a hub of early facial ectomesenchyme specification. Interestingly, unlike Osteichthyes, where the frontonasal ectomesenchyme rapidly accumulates in the anterior part of the face, in the catshark, the cells remain confined to the periocular region (Fig. 5).

To assess whether the *TWIST1*⁺ periocular ectomesenchyme is homologous to the frontonasal mesenchyme observed in Osteichthyes, we examined the expression of positional identity genes *ALX1* and *DLX2*, markers of the frontonasal mesenchyme and pharyngeal arches, respectively (Square et al., 2015, 2017). *ALX1* was expressed in the head mesoderm at St.16-18 (Fig. 5B and Fig. S9) and subsequently detected in the periocular ectomesenchyme from St.19 onward, extending toward the olfactory placode by St.25. *DLX2* was primarily expressed in the pharyngeal arch ectomesenchyme from St.19 onwards. Notably, there is a brief developmental window at St.19-20 when *ALX1* and *DLX2* colocalize in the periocular mesenchyme, with *ALX1* becoming dominant by St.21. In amniotes, early expression of *DLX2* is also temporarily observed in the periocular ectomesenchyme, which will give rise to the maxillary prominence (Lee et al., 2004). Whether the *DLX2*⁺ periocular mesenchyme in the catshark contributes the maxillary prominence remains to be determined. Nonetheless, the robust *ALX1* expression in the catshark periocular mesenchyme suggests that it may be equivalent to the frontonasal ectomesenchyme found in other Osteichthyes.

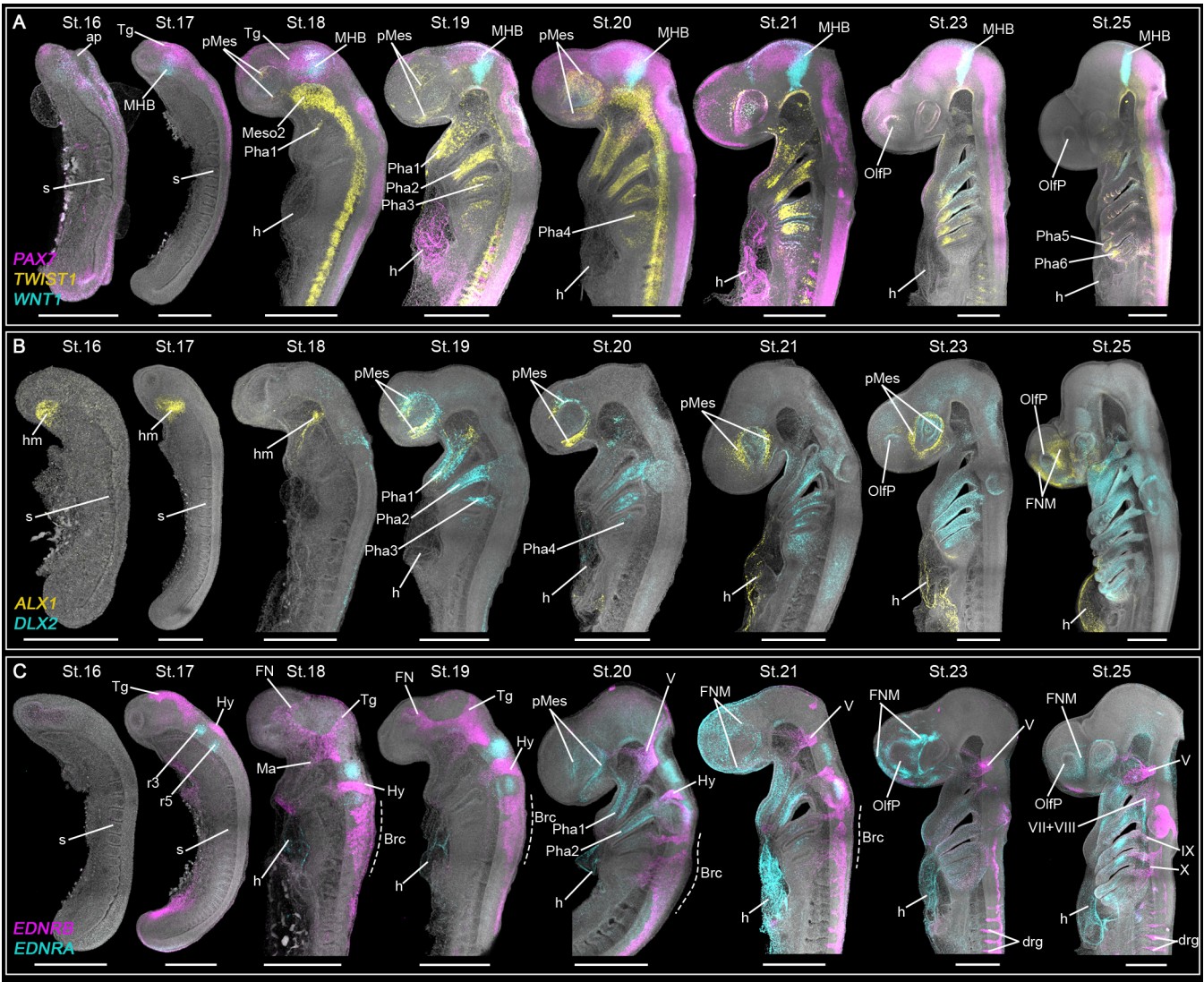

**Fig. 5. Mesenchymal gene expression patterns during catshark embryogenesis.** Expression patterns of *PAX7*, *TWIST1* and *WNT1* (A), *ALX1* and *DLX2* (B), and *EDNRB* and *ENDRA* (C). Scale bars: 500 μm. ap, anterior neuropore; Brc, branchial CNCC; drg, dorsal root ganglia; FN, frontonasal branch; FNM, frontonasal ectomesenchyme; h, heart; hm, head mesoderm; Hy, hyoid CNCC; IX, glossopharyngeal ganglia; Ma, mandibular branch; Meso2, mesoderm 2; MHB, midbrain-hindbrain boundary; OlfP, olfactory placode; Pha1-6, first to sixth pharyngeal arches; pMes, periocular ectomesenchyme; r3-5, third and fifth rhombomeres; s, somite; Tg, trigeminal CNCC; V, trigeminal ganglia; VII+VIII, acousticofacial ganglia; X, vagus ganglia.

To further explore the evolutionary conservation of facial patterning signals, we investigated the endothelin (EDN) pathway (Fig. 5C and Fig. S10), identified by CellChat as a strong signalling input to ectomesenchymal cells (Fig. 3F). In model Osteichthyes and cyclostomes, *EDNRA* has been linked to adaptive skeletal phenotypes, while *EDNRB* is involved in pigmentation and PNS development (Square et al., 2016, 2020). In the catshark, as observed in other vertebrates, *EDNRB* is expressed in early CNCCs and becomes restricted to the PNS by St.21 (Fig. S10A,B). In contrast, *EDNRA* is expressed in the facial ectomesenchyme, both pharyngeal and periocular, starting at St.20 (Fig. 5C). The conserved expression patterns of the EDN receptors in the catshark suggest functional conservation of this signalling pathway across vertebrates. *EDNRA* expression in the periocular ectomesenchyme, despite its lateral position within the periocular region, further suggests the conserved role of this mesenchyme in skeletogenic patterning across vertebrates. However, further functional validations will be required to assess the presumed conserved role of the EDN pathway in Chondrichthyes.

Together, these data reveal the distinct spatial and temporal organization of the facial ectomesenchyme in the catshark. Our results demonstrate that, despite its divergent anatomical position, the catshark periocular ectomesenchyme bears key molecular features of the frontonasal ectomesenchyme observed in Osteichthyes, suggesting its homology. Cells from this region appear to be the cellular source of the catshark face. Despite the divergent anatomical position, the catshark periocular ectomesenchyme retains conserved positional markers, reinforcing the functional homology of these ectomesenchymal domains.

### SRµCT reconstruction of catshark facial morphogenesis

The facial ectomesenchyme gives rise, among other tissues, to cartilage, which forms a blueprint for the skeletal elements of the face and dictates its shape. Cartilage formation begins with the localized intensive proliferation of ectomesenchymal cells, known as mesenchymal condensations. The shape of the mesenchymal condensations forecasts the general geometrical blueprint of the

future skull (Kaucka et al., 2017). While disruptions in the development of mesenchymal condensations result in a spectrum of craniofacial malformations (Kaucka et al., 2017), during evolution, differences in their positioning and growth generate the spectrum of facial geometries found in vertebrates (Kyomen et al., 2023). To understand craniofacial morphogenesis in the catshark, we reconstructed the developmental trajectory of the ectomesenchyme using synchrotron-radiation micro-computed tomography (SRμCT) (Fig. 6).

We first examined CNCC migration and ectomesenchyme formation at St.18-25 (Fig. 6), and focused on the trigeminal CNCC population (frontonasal and mandibular branches), which collectively build the entire catshark face. Three-dimensional (3D) reconstructions of CNCCs at St.18 revealed migratory streams consistent with observations from SOX9 immunofluorescence and gene expression mapping, including a distinctive frontonasal branch that follows the head midline and populates the upper periocular region (Figs 1, 5 and 6A). By St.19, CNCCs reach the lower periocular region and begin to surround the optic grooves. The continued proliferation and expansion of the upper and lower periocular ectomesenchyme results in their convergence by St.21, forming a single periocular ectomesenchyme (Fig. 6A). Frontal view and 3D reconstructions revealed that the right and left facial ectomesenchymal domains do not merge and remain as two bilateral domains (Fig. 6B). Instead, the midfacial ectoderm, which is located in the anterior most of the face in between the nasal placodes, is still in direct contact with the forebrain (Fig. 6C,D). Fusion of the left and right periocular ectomesenchymal domains into a continuous bona fide frontonasal ectomesenchyme layer effectively separates the brain from the surface ectoderm at St.25 (Fig. 6A,D). This stage also marks the emergence of a clear division between the mandibular (MdP) and maxillary (MxP) prominences.

The process of CNCC migration and facial ectomesenchyme formation in the catshark takes ∼35 days (St.18-25), in contrast to the 3-day process in mouse embryos [embryonic days (E)8.5-10.5]. Despite the temporal disparity, the sequence of developmental events is highly conserved. In the mouse model, CNCC specify, delaminate and migrate by E8.5, reaching the anterior most of the face directly within 24 h. By E9.5, the early frontonasal ectomesenchyme is established as bilateral domains that are still separated by the midfacial ectoderm (Fig. 6E), similar to the lateral segregation observed in the catshark at St.20-21 (Fig. 6A,B). Notably, the mouse frontonasal ectomesenchyme does not surround the eye region, as in the catshark, and instead accumulates in a more-rostral position. Furthermore, in mouse, the early frontonasal ectomesenchyme undergoes rapid proliferation and outgrowth, forming the facial prominences by E10.5 (Fig. 6E). This process coincides with the invagination of the olfactory placode, leading to the emergence of the lateral and medial nasal prominences (the LNP and MNP) at E10.5. In contrast, the catshark does not exhibit comparable growth of the frontonasal ectomesenchyme to form the nasal prominences. Hence, invagination of the catshark nasal placode at St.23-25 results in anterior and posterior ectomesenchymal domains (the AED and PED), whose homologies to the mouse LNP and MNP remain unresolved. However, based on the general developmental parallels (invagination of nasal placode, initial growth of the MxP, uniform bona fide frontonasal ectomesenchyme, prominent forebrain and well-established eyes without pigment), we propose that St.25 in the catshark is equivalent to E10.5 in mouse.

We next analysed later stages of catshark development (St.27-31), during which embryos undergo considerable morphological changes (Fig. 7). At St.27, the MxP shows a notable growth, while the overall facial ectomesenchyme architecture remains similar to St.25 (Figs 6A, 7). The growth of the MxP precedes the transformation of the mouth from an embryonic diamond shape to the characteristic oval morphology of the adult. By St.28, coinciding with the mouth re-shaping, we observed the earliest mesenchymal condensations, most prominently in the upper and lower jaw regions, corresponding to the future palatoquadrate and Meckel's cartilage, respectively. We tracked the growth of these mesenchymal condensations across subsequent stages (St.29-31) (Fig. 7) and validated the accuracy of the segmented STμCT 3D models by SOX9 whole-mount immunofluorescence (Fig. S11). These condensations, along with the cartilaginous elements derived from them, were clearly identifiable in STμCT tomographic slices (Fig. S12; see Materials and Methods). By St.30, condensations corresponding to the rostrum and the nasal capsule were also visible, preceding the initial facial protrusion at St.31 (Ballard et al., 1993). Notably, the shape of the early mesenchymal condensations closely matched the morphologies of their resulting cartilaginous elements, supporting the notion that the chondrogenic condensations are spatially pre-patterned prior to chondrocyte differentiation (Kaucka et al., 2017). Importantly, in Chondrichthyes, the cranial mesoderm contributes to the formation of the posterior pharyngeal skeleton, but does not contribute to the jaw cartilage (Sleight and Gillis, 2020). Hence, the distinct cartilaginous elements studied in our SRμCT analysis, which derive from the trigeminal CNCCs, are likely of CNCC origin, without any mesodermal contribution.

## DISCUSSION

The evolution of CNCCs in gnathostomes marked a major evolutionary innovation that contributed to the success and diversification of this lineage (Kyomen et al., 2023; Martik and Bronner, 2021; Martik et al., 2019; Stundl et al., 2020; Wakamatsu et al., 2014). To date, research on model osteichthyans yielded fundamental knowledge on CNCC developmental trajectories, together with their underlying molecular programmes and GRN. However, a significant knowledge gap remains regarding these processes in Chondrichthyes. This incomplete taxon sampling hinders reconstructing the ancestral CNCC programmes, identifying sources of developmental divergence and understanding the role of CNCC in lineage-specific adaptations. Here, by leveraging the catshark as a representative of Chondrichthyes, the sister group to Osteichthyes, we reveal the cellular and molecular underpinnings of catshark-specific CNCC biology and early facial morphogenesis. We demonstrate that CNCC development in the catshark is largely conserved compared to other gnathostomes; however, lineage-specific adaptations within the ectomesenchymal derivatives provide insights into the evolutionary diversification of the craniofacial architecture in Chondrichthyes (Fig. 8A).

CNCCs delaminate following neural tube closure in the catshark and the little skate (Sleight and Gillis, 2020). However, the timing of CNCC delamination varies across vertebrate lineages, which hinders the reconstruction of the ancestral condition. In mammals, amphibians and basal actinopterygians (bichir, sterlets and gars), CNCC delaminate prior to neural tube closure (Stundl et al., 2020; Theveneau and Mayor, 2012; Wakamatsu et al., 2014). In contrast, delamination after neural tube closure is observed in saurians (birds, crocodiles, squamates and turtles), lungfishes, Chondrichthyes and cyclostomes (Diaz et al., 2019; Ericsson et al., 2008; Kundrát, 2009; Square et al., 2017; Theveneau and Mayor, 2012). Although this pattern of CNCC delamination appears phylogenetically scattered, the shared post-closure delamination among cyclostomes, Chondrichthyes and key osteichthyan lineages suggests that delamination after neural tube closure is the ancestral

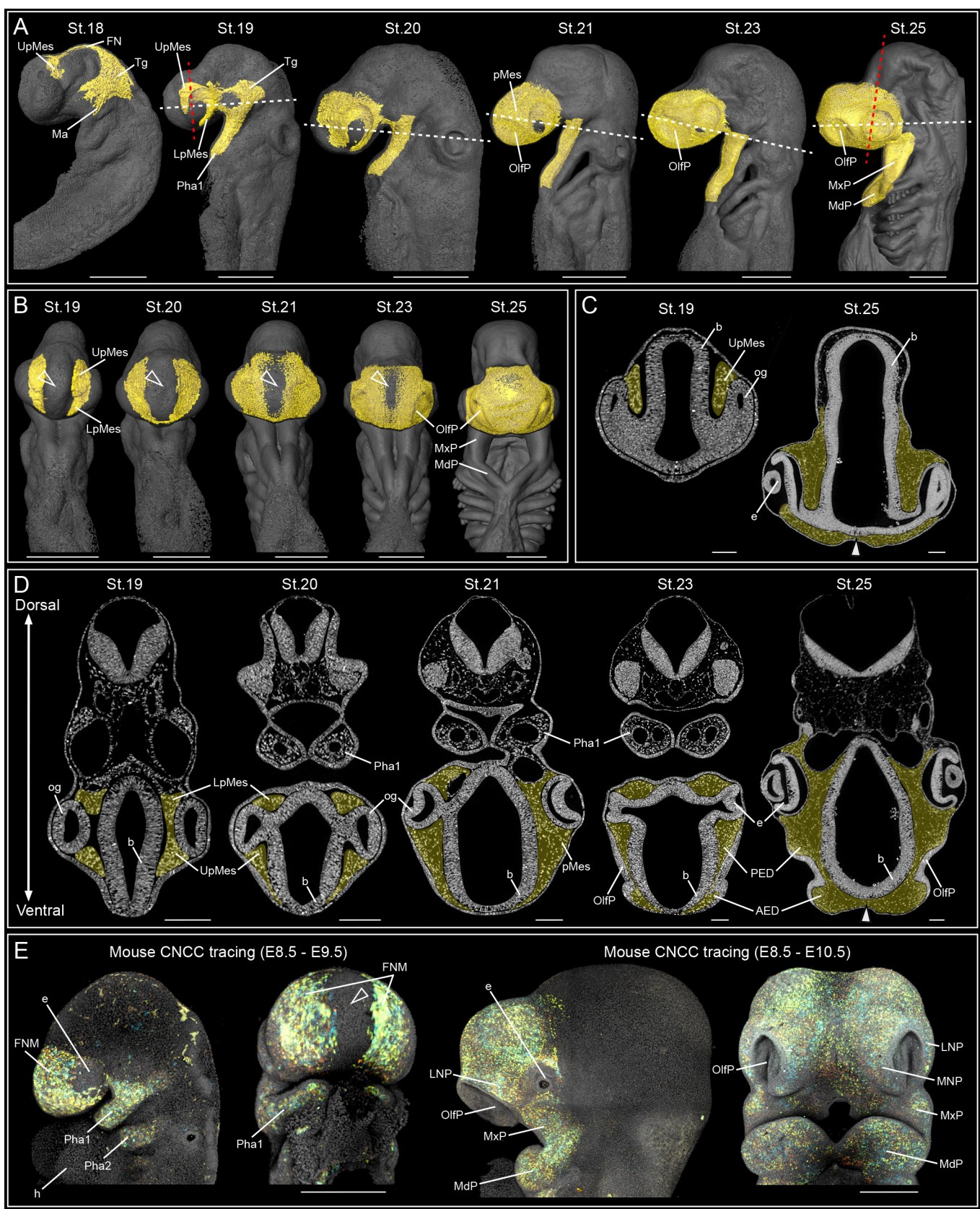

**Fig. 6.** See next page for legend.

vertebrate state. Consequently, delamination prior to neural tube closure likely represents a derived condition that evolved independently multiple times within Osteichthyes.

Catshark CNCC migrate via the canonical trigeminal, hyoid and branchial streams, resembling the migratory patterns observed in other vertebrate species (Minoux and Rijli, 2010; Stundl et al., 2020;

**Fig. 6. 3D reconstruction of the catshark facial ectomesenchyme.**
(A,B) Lateral (A) and ventral (B) views of reconstructed CNCCs (St.18-19) and early facial ectomesenchyme (St.20-25). Only the frontonasal ectomesenchyme is shown in B. Arrowheads in B indicate the midfacial ectoderm. Dashed red and white lines indicate the positions of the tomographic slices shown in C and D. Scale bars: 500 μm. (C) Tomographic slices highlighting the position of CNCCs in the supraocular region at St.19 and facial ectomesenchyme at St.25. White arrowhead indicates a thin mesenchymal domain separating the ectoderm from the brain. Scale bars: 100 μm. (D) Tomographic slices highlighting the position of CNCCs and facial ectomesenchyme. White arrowhead indicates the first signs of a uniform frontonasal ectomesenchyme at St.25. Note the invagination of the olfactory placode divides the frontonasal ectomesenchyme into an anterior (AED) and posterior (PED) domain by St.25. Scale bars: 100 μm.
(E) Genetic tracing of mouse CNCCs and their ectomesenchymal derivatives induced at E8.5 in *Sox10-creERT2/R26Confetti* embryos and analysed at E9.5 (left) and E10.5 (right). Embryos are shown in lateral and frontal views. Scale bars: 500 μm. White-outlined arrowheads indicate the midfacial ectoderm. AED, anterior ectomesenchymal domain; b, brain; e, eye; FN, frontonasal branch; FNM, frontonasal ectomesenchyme; h, heart; LNP, lateral nasal prominence; LpMes, lower periocular ectomesenchyme; Ma, mandibular branch; MdP, mandibular prominence; MNP, medial nasal prominence; MxP, maxillary prominence; og, optic groove; OlfP, olfactory placode; PED, posterior ectomesenchymal domain; Pha1-2, first and second pharyngeal arches; pMes, periocular ectomesenchyme; Tg, trigeminal CNCC; UpMes, upper periocular ectomesenchyme.

Theveneau and Mayor, 2012) (Fig. 8A,B). Catshark CNCC arise primarily from the hindbrain region, with limited contribution from the caudal midbrain. Notably, we did not observe CNCC migrating from the forebrain region. This is similar to the pattern seen in the sea lamprey, where only the caudal midbrain contributes to the trigeminal stream, and forebrain-derived CNCC have not been reported to date (McCauley and Bronner-Fraser, 2003; Square et al., 2017). In Actinopterygii (ray-finned fishes), the rostral midbrain additionally

contributes to the CNCC trigeminal stream but no forebrain-derived CNCCs have been detected (Kwak et al., 2013; Rocha et al., 2020). Given that the presence of forebrain-derived CNCCs has only been observed in Sarcopterygii (Couly et al., 2002; Epperlein et al., 2000; Le Douarin, 2004; Martik and Bronner, 2021; Serbedzija et al., 1992), this population may represent a later evolutionary expansion of the ancestral cranial domain (Fig. 8A). This hypothesis is further supported by previous work on the cloudy catshark (*Scyliorhinus torazame*) (Kuratani and Horigome, 2000). However, further experimental evidence, ideally employing lineage tracing across Chondrichthyes species, will be required to confidently rule out the existence of rostral midbrain- and forebrain-derived CNCC in Chondrichthyes and mark them as a Sarcopterygii-specific novelty (Wen et al., 2026).

Various Actinopterygii lineages, such as bichirs, gars and certain Teleost species (e.g. northern pike), exhibit an accelerated development of the hyoid CNCC stream relative to the trigeminal stream, which is associated with the early formation of gill-related structures (Stundl et al., 2020). In contrast, in Sarcopterygii, the trigeminal stream is the most prominent, and the first one to start migration and reaching its target destination, while the hyoid stream is usually smaller (Minoux and Rijli, 2010; Theveneau and Mayor, 2012). Our analysis of catshark CNCCs, revealed a pattern consistent with that observed in Sarcopterygii, with the trigeminal stream being the most prominent and advanced CNCC population. These findings suggest that the accelerated development of the hyoid CNCCs in actinopterygiians represents a derived feature due to a heterochronic shift, and that the ancestral gnathostome state likely resembled the pattern seen in sarcopterygians and the catshark (Fig. 8A,B).

Chondrichthyes display an impressive variety of facial morphologies: sharks of the hammerhead family present with flattened and laterally expanded heads; sawfish have a characteristic

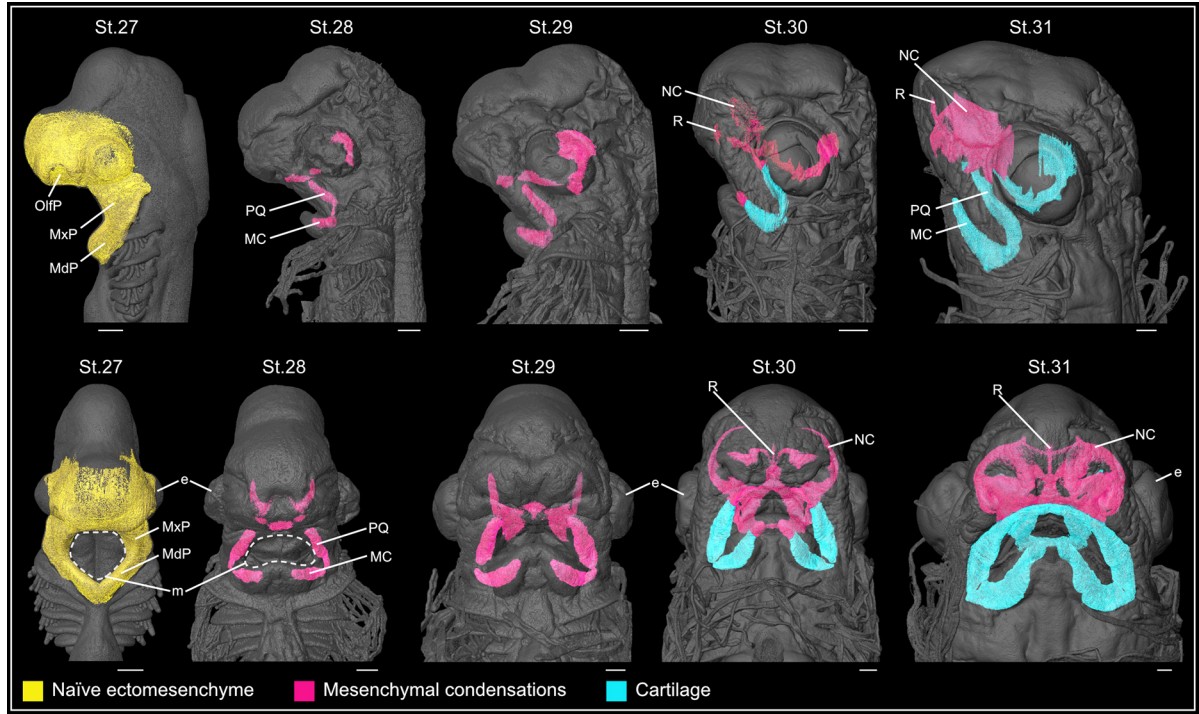

**Fig. 7. 3D reconstruction of the catshark facial chondrocranium.** Lateral (top) and ventral (bottom) view of naïve facial ectomesenchyme (St.27), mesenchymal condensations and cartilage (St.28-31) during catshark embryogenesis. The basic shape of the chondrocranium is already established as early as St.28. Scale bars: 500 μm. e, eye; m, mouth; MC, Meckel's cartilage; MdP, mandibular prominence; MxP, maxillary prominence; NC, nasal capsule; OlfP, olfactory placode; PQ, palatoquadrate; R, rostrum.

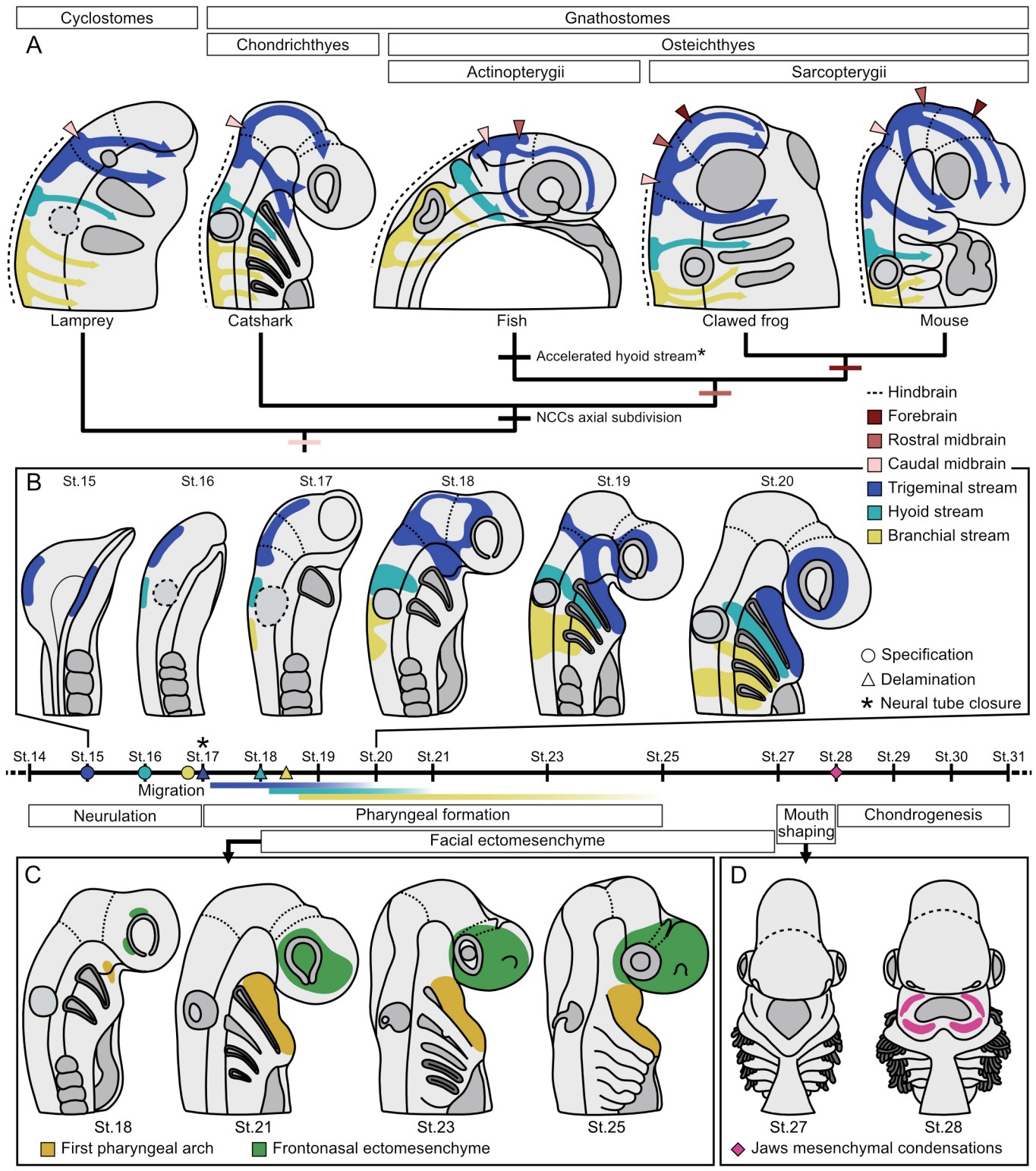

**Fig. 8. Evolutionary perspective and schematic summary.** (A) Proposed model of CNCC evolution. Arrow thickness indicates the most prominent stream. Coloured arrowheads indicate the anatomical location within the CNS where trigeminal CNCC are specified. Only forebrain-derived CNCCs are present in Sarcopterygii. The embryonic Actinopterygii schematic representation is a prototypical fish embryo. Black asterisk indicates that, while certain actinopterygian lineages like bichirs, gars and pikes display an accelerated hyoid stream, this phenomenon has not been observed in other species like sterlets and zebrafish (Stundl et al., 2020). Hence, the accelerated hyoid stream is a feature of certain actinopterygian lineages. It is currently unknown whether this feature is present in other species or represents the ancestral state of the entire clade. (B) Schematic representation of CNCC development in the catshark (top) and timeline of CNCC-related events during catshark development (bottom). (C) Positioning of the facial ectomesenchyme in the catshark. (D) Re-shaping of the mouth during catshark development coincides with the appearance of the first mesenchymal condensations. Only jaw mesenchymal condensations are depicted.

elongated frontal rostrum with sharp transverse teeth; manta rays feature cephalic lobes located on either side of the mouth; and elephant sharks (*Callorhinchus milii*) possess a distinctive flexible snout. These facial shapes are ultimately dictated by the internal skeletal elements forming the skull (Byrum et al., 2024; Kaucka et al., 2017; Pitirri et al., 2020), primarily of CNCC origin (Martik and Bronner, 2021; Santagati and Rijli, 2003). Thus, the immense variability of facial morphologies in Chondrichthyes, and potentially in all gnathostomes, may be to a large extent attributed to the evolution of CNCCs and, in particular, their mesenchymal programme. The presented scRNA-seq dataset and gene expression mapping jointly indicate that the transcriptional signature of CNCCs and the early ectomesenchyme-specifying molecular programme are largely conserved between Osteichthyes and Chondrichthyes, (Adhikari et al., 2025; Compagnucci et al., 2013; Square et al., 2015, 2016, 2020). However, the spatiotemporal dynamics of the formation and position of the frontonasal ectomesenchyme strikingly differs compared to Osteichthyes.

Unlike in Osteichthyes, where CNCCs migrate rapidly to the anterior part of the face, CNCCs in the catshark first accumulate around the periocular region, where they activate their mesenchyme-specifying transcriptional programme (Fig. 8C). This periocular ectomesenchyme aggregation has been observed in other chondrichthyans, including the little skate (Sleight and Gillis, 2020), elephant shark (Johanson et al., 2015) and cloudy catshark (Kuratani and Horigome, 2000), suggesting a lineage-specific trait. Despite this positional divergence, based on the expression of conserved positional genes such as *ALX1* and *ALX4* (Compagnucci et al., 2013), we propose that the chondrichthyan periocular ectomesenchyme is homologous to the early frontonasal mesenchyme in Osteichthyes. While lineage-tracing studies are needed to definitively confirm the contribution of periocular mesenchyme to rostral cranial elements in Chondrichthyes, our findings suggest that the rostral region of the facial skeleton, like the rostrum and nasal capsule, is derived from the elusive frontonasal ectomesenchyme. In summary, this positional divergence of periocular, frontonasal-like ectomesenchyme may have functional implications for the shaping of the chondrocranium and could underlie the lineage-specific facial morphologies observed in Chondrichthyes (Fig. 8C). Future comparative analyses incorporating additional chondrichthyan species with divergent facial morphologies will be essential to assess the broader applicability of our findings from the small-spotted catshark and to further evaluate the functional significance of the periocular ectomesenchyme behaviour. At present, however, extending evolutionary developmental studies beyond the small number of experimentally accessible chondrichthyan species remains a major challenge, reflecting the non-model status and limited embryonic availability of most members of this clade.

Despite the extensive facial diversity among adult chondrichthyans, embryos exhibit striking morphological similarities up to postpharyngeal development, when they begin to acquire the species-specific traits (Ballard et al., 1993; Byrum et al., 2024; Didier et al., 1998; Gillis et al., 2022; Onimaru et al., 2018; Rodda and Seymour, 2008; Vazquez et al., 2022). This developmental window corresponds to the reshaping of the mouth from its embryonic diamond shape to the adult morphology and the appearance of the first mesenchymal condensations (Gillis et al., 2009, 2012). The mesenchymal condensations of the pharyngeal skeleton have been previously mapped in the catshark using traditional histochemical staining in sections (Gillis et al., 2009, 2012), which does not allow the complex three-dimensional geometry of these skeletal elements to be determined. Our SRμCT analysis and 3D reconstructions confirmed the presence of early mesenchymal condensations in the jaws at St.28

and the onset of chondrogenesis by St.30-31 (Gillis et al., 2009, 2012; Grogan et al., 1999; López-Romero et al., 2022) (Fig. 8D). Moreover, we show that, similar to mouse, the mesenchymal condensations in the catshark are induced spatially and geometrically patterned (Kaucka et al., 2017). This suggests that facial cartilage patterning is defined early during mesenchymal commitment and aggregation, with mesenchymal condensations being outputs of these early developmental processes, acting as developmental blueprints for species-specific facial morphology. Furthermore, our data support the hypothesis that the genetic programmes driving the induction and formation of the mesenchymal condensations are likely shared across gnathostomes.

An advantage of the catshark as a model organism lies in its long embryonic development (~175 days) (Ballard et al., 1993), which allows the capture of the intermediate and transitioning states of complex developmental processes that are otherwise challenging to obtain in conventional fast-developing models, such as mouse and chicken (roughly 21 days). For instance, in mouse and zebrafish, early-migrating CNCCs give rise predominantly to ectomesenchymal cells, while late-migrating CNCCs tend to generate neuroglial cell types (Erickson et al., 2024 preprint; Tatarakis et al., 2021). Due to the rapid development in these species, the temporal separation between these two cell lineages occurs within a narrow interval of ~12 h. In contrast, we observe a similar fate bifurcation over 7 days in the catshark, spanning St.18-22. This extended timeframe offers a valuable window to investigate the gene regulatory shifts underpinning CNCC fate decisions with high temporal precision.

Future studies will be required to further refine the developmental dynamics of CNCCs and facial morphogenesis in the catshark. Due to the difficulty in obtaining sufficient numbers of embryos at individual early developmental stages, a limitation of the present work is that the single-cell dataset was generated from pooled embryos spanning multiple developmental stages. This limits precise assignment of transcriptional states to individual timepoints, although stage-specific expression mapping to a large extent mitigates this issue. Extending single-cell profiling to later embryonic stages will be important to achieve finer temporal resolution of CNCC differentiation, particularly with respect to the emergence of neuroglial derivatives, which likely arise after the developmental window analysed here. In addition, complementary lineage-tracing approaches would enhance our understanding of CNCC trajectories and cellular behaviours. Finally, gene-specific functional validation, through approaches such as CRISPR-mediated mutagenesis or morpholino-based knockdown, would provide crucial insight into CNCC biology and ectomesenchymal dynamics. At present, however, targeted genetic manipulation has not yet been established in chondrichthyan embryos, particularly during early development, and therefore remains beyond the scope of the current study. In this context, the integrative molecular, spatial and morphological framework provided here offers a necessary reference and foundation for future functional and comparative studies of NCC development in this phylogenetic node.

This study establishes a comprehensive and information-rich platform for future research on shark embryogenesis and the development and evolution of CNCCs. The accompanying open-access, interactive single-cell transcriptomic and microscopy atlases allow users to explore, visualize and interrogate the datasets presented at www.evolbio.mpg.de/escamilla-sharknc. Although further research is needed to fully reconstruct the complete temporal spectrum of CNCC fate commitments in the catshark, this study provides new insights into both conserved and lineage-specific aspects of gnathostome craniofacial development. Through

the integration of scRNA-seq, spatial gene expression mapping and SRμCT-based three-dimensional reconstructions, we show that while core CNCC molecular programmes are broadly conserved across jawed vertebrates, the spatial deployment and cellular behaviour of the facial ectomesenchyme differ in key ways between Chondrichthyes and Osteichthyes. Notably, these differences may underlie lineage-specific facial architectures and provide a developmental basis for the morphological diversity observed across gnathostomes. Importantly, our findings highlight the value of the small-spotted catshark as a representative chondrichthyan species in evolutionary developmental biology. As such, expanding research into non-model organisms from underexplored lineages is essential to uncover the full evolutionary and developmental context of vertebrate facial diversity and enable reconstruction of ancestral conditions and derived traits.

## MATERIALS AND METHODS
### Animal information
All animal work was conducted following Directive 2010/63/EU, the German Animal Welfare Act (Tierschutzgesetz § 11) and in compliance with the Federation of European Laboratory Animal Science Associations' (FELASA) international and institutional guidelines for the housing, handling and euthanasia of laboratory animals. The collection of small-spotted catshark (*Scyliorhinus canicula*) and chicken (*Gallus gallus*) embryonic stages used in this study does not require an ethical permit and short-term housing of the eggs was approved by the local veterinary officer (Veterinäramt Kreis, Plön). Fertilized catshark eggs were provided by Martin Hansel (Sea Life Berlin, Berlin, Germany) and also obtained from two breeding groups at Ozeaneum (Deutsches Meeresmuseum, Stralsund, Germany) (details below). Fertilized chicken eggs were purchased from LOHMANN Deutschland (Ankum, Germany). All embryology work performed in this study was carried out following the current international animal guidelines and in compliance with the German Animal Welfare Act [§ 4 (3) TierSchG].

The C57Bl/6J mouse strain was purchased from Charles River Laboratories Germany (code 632). The *Sox10-CreERT2* mouse strain was produced by the laboratory of Dr Vassilis Pachnis and has been previously described (Laranjeira et al., 2011). The *Sox10-CreERT2* line was coupled with the *R26Confetti* mouse strain (The Jackson Laboratory, code 013731) to generate *Sox10-CreERT2/R26Confetti* mice. Mice were housed at the Max Planck Institute for Evolutionary Biology's Animal facility. The housing and maintenance of mice are approved by the MLLEV (Ministerium für Landwirtschaft, ländliche Räume, Europa und Verbraucherschutz des Landes Schleswig-Holstein) under permit PLÖ-0004697. The collection of mouse embryonic stages during the first and second trimesters of gestation was performed according to the German Animal Welfare Act [§ 4 (3) TierSchG] and reported to the Animal Welfare Officers (Kiel, Germany) under permit number 1398. Genetic tracing of neural crest cells using the *Sox10-CreERT2/R26Confetti* line is approved under permit 43-6/21.

### Small-spotted catshark husbandry and egg collection
Adult catsharks housed at Ozeaneum (Deutsches Meeresmuseum, Stralsund) were divided in two breeding groups consisting of five females and one male each. Both breeding groups were kept in 3800 L recirculating artificial sea water tanks at 14-15°C and 33-35 PSU salinity where they can mate *ad libitum*. Catsharks display an aggressive mating behaviour in which males bite and hold the pectoral fins of the female to place them into mating position; hence, females were checked daily for extensive injuries and males were removed from the tank to allow females to recover for 1-2 weeks when needed. Catsharks housed at Sea Life Berlin (Berlin, Germany) are part of a display tank and were kept under similar conditions. After mating, female catsharks deposit the eggs around an artificial kelp positioned in the middle of the tanks. Eggs were collected once a week from the breeding tanks by carefully removing them from the kelp and transferring to a mesh bag that was then placed in a separate tank connected to the same circulation system

as the breeding tanks. Eggs were checked on a regular basis by flashing them with a light source ('candling') to visualize their content. Eggs were considered healthy if they had an intact, ellipsoid yolk inside. When an egg has spoiled, it is easily identified due to the bad state of the yolk, which can be flowing inside the egg case and detaching from the vitelline membrane. Moreover, 'candling' also allows the developing embryo to be visualized within the egg. Upon reaching a certain embryonic stage, embryos will start moving, which can be used to identify dead embryos due to the lack of movement. Unhealthy eggs and eggs containing dead embryos were immediately removed from the tanks and discarded from future experiments. Once enough eggs were collected, they were shipped to the facilities of the Max Planck Institute for Evolutionary Biology, where they were maintained in oxygenated sea water at 17-18°C until the required embryonic stages were reached (Ballard et al., 1993).

### Embryo collection and fixation
Catshark eggs were opened using sharp dissecting scissors, taking care not to damage the yolk and the whole content of the egg (jelly, yolk and embryo) was placed in a glass Petri dish with sea water. The jelly inside the egg should be transparent; in unhealthy eggs it becomes murky. The jelly was removed to better visualize the embryos, which were then carefully dissected under a stereomicroscope and staged according to Ballard (Ballard et al., 1993). The embryos were euthanized by a tricaine (ethyl-3-aminobenzoat-methansulfonat, Merck, E10521) overdose and fixed in freshly made 4% paraformaldehyde (PFA) in 1×Dulbecco's phosphate-buffered saline (PBS) (Sigma D5652) at 4°C for 1-4 h on slow rotation, depending on the developmental stage and experimental application. For whole-mount immunofluorescence staining and *in situ* hybridization, embryos were dehydrated in an increasing methanol series for 10 min each (25%, 50%, 75% and 100%) in 0.1% PBS-Tween (PBST) (Tween 20; Sigma P9416) at 4°C on a slow rotation and stored in methanol at −20°C until further use. Catshark embryos for single-cell transcriptomics were prepared according to the protocol described below.

Mice mating pairs were set overnight and the presence of a vaginal plug was assessed the following morning. Noon of the plug-positive day was considered embryonic day E0.5. Upon reaching the desired embryonic stage (E8.5-E9.5), pregnant mice were euthanized by cervical dislocation, and embryos were dissected in sterile ice-cold PBS under a stereomicroscope. The collection of mouse embryos for genetic tracing of CNCCs using the *Sox10-creERT2/R26Confetti* is described below. Fertilized chicken eggs were kept in a humidified rotating incubator at 37°C. Upon reaching the desired Hamburger-Hamilton (HH) stage (HH12-HH16) (Hamburger and Hamilton, 1992), the embryos were carefully dissected from the yolk and washed in ice-cold PBS under a stereomicroscope. Mouse and chicken embryos were fixed and processed as described above for small-spotted catshark embryos.

### Sample preparation for single-cell transcriptomics
After removing the catshark embryos from the egg and placed in a glass Petri dish, they were anesthetized on ice for 10 min, which reduces their movements and allows for fast and proper staging based on morphological landmarks defined by Ballard and colleagues (Ballard et al., 1993). In order to capture only the CNCCs, the head region was cut at the level of the otic capsule with fine dissecting scissors and forceps, and placed in ice-cold sea water on ice during the whole duration of the staging and dissecting process, to reduce cell mortality and maintain osmolarity. Heads coming from embryos of the following stages were dissected and pooled together in one single tube: St.16 ($n=3$), St.17 ($n=2$), St.18 ($n=1$), St.19 ($n=2$) and St.20 ($n=1$). Once all embryonic heads were dissected, they were centrifuged at 500 $g$ for 5 min at 4°C and sea water was carefully removed. To remove any traces of salt that could impair future enzymatic steps, heads were washed with ice-cold PBS and centrifuged at 500 $g$ for 5 min at 4°C. After carefully removing the PBS, heads were incubated in 1 ml of pre-warmed trypsin solution (0.05% Trypsin/0.02% EDTA – Pan Biotech P10-0235SP) at 37°C for 10 min with gentle shaking. During trypsin incubation, samples were gently vortexed for 30 s every 5 min to ensure a thorough mixing. All steps after trypsin treatment were performed inside a clean bench to avoid ambient contamination. Following trypsinization, heads were dissociated by gently pipetting five times with a P1000, resulting in a homogeneous cell

suspension. The trypsin reaction was quenched by adding 200 µl of pre-warmed FBS (Sigma F0804). Cells were centrifuged at 500 *g* for 5 min at 4°C and the supernatant removed without disturbing the cell pellet. Cells underwent three washes in 1 ml sterile-filtered ice-cold PBS supplemented with 2.5 µl of RNase inhibitors (Thermo Fisher Scientific N8080119) at 500 *g* for 5 min at 4°C to remove any traces of trypsin and cell debris. The resulting cell suspension was filtered through a 35 µm cell strainer (Falcon 352235) to remove cell clumps and obtain a final single-cell suspension. Cell viability and concentration was calculated by staining an aliquot of the cell suspension with 0.4% Trypan Blue (Gibco 15250061) in a 1:1 proportion, loading it to a C-Chip disposable haemocytometer slide (NanoEnTek DHC-N01) and assessed using a Leica DMLS light microscope (020-518.500). The sample had a final cell viability of around 95% and cell concentration of 650 cells/µl.

### scRNA-seq library preparation and sequencing

10,000 cells were targeted following the 10x Genomics protocol for the Chromium platform using the Chromium Next GEM Single Cell 3′ Reagent Kits v3.1 Dual Index (10x Genomics PN-1000269). All subsequent steps (GEMs generation, library preparation and sequencing) were performed in line with the manufacturer's guidelines. GEMs were obtained using the Chromium Next GEM Single Cell 3′ GEM Kit v3.1 (10x Genomics PN-1000130) and Gel Beads Kit v3.1 (10x Genomics PN-1000129). The scRNA-seq library was generated using the Chromium Next GEM Single Cell 3′ Library Kit v3.1 (10x Genomics PN-1000196) and quantified using the Invitrogen Qubit dsDNA assay-kits (FisherScientific 10616763) in a fluorospectrometer (ThermoFisher ND-3300). Sequencing was performed by Novogene in a NovaSeq 6000 platform (PE150bp strategy), which generated around 50,000 reads per cell.

### scRNA-seq bioinformatic data analyses

The 10x Genomics Cell Ranger v7.1.0 pipeline was used to process the scRNA-seq data. Briefly, raw sequencing files were demultiplexed and converted to FASTQ format using 'cellranger mkfastq'. A custom catshark reference genome was constructed based on the catshark genome assembly sScyCan1.1 (GCA_902713615.1) using the 'cellranger mkred' function. Reads were then aligned to the catshark custom reference genome and quantified by 'cellranger count' to produce a raw count matrix. The raw count matrix was further processed in Python v3.9.16 using Scanpy v1.9.3 (scverse.org) and anndata v0.10.3. Quality control metrics were calculated using the 'pp.calculate_qc_metrics()' function. The raw count matrix was filtered to only keep cells with 1000-11,000 genes, 7500-100,000 counts, 0.1-5.5% mitochondrial content and 5-25% ribosomal content. Filtering based on the previous parameters resulted in 6585 cells with an average depth of 28,807 reads and 5760 genes per cell. The raw count matrix was normalized and log transformed using 'pp.normalize_total()' and 'pp.log1p()', respectively. Initial dimensionality reduction was performed using a Principal Component (PC) Analysis (PCA) on the top highly variable genes identified using 'pp.highly_variable_genes()' function with the following settings: 'min_mean=0.0125, max_mean=3, min_disp=0.5'. Further dimensionality reduction was performed on the first 25 PCs using the Uniform Manifold Approximation and Projection (UMAP) algorithm to embed cells in a two-dimensional plane. Cells were clustered using the 'tl.leiden()' function with resolution 0.5. Once the neural crest cells were identified, they were extracted from the main dataset and re-clustered with resolution of 0.3. To assist in the cluster annotation process, differentially expressed genes (DEGs) were calculated using the 'tl.rank_genes_groups()' with the Wilcoxon method.

### Cell cycle state prediction

Catshark cell cycle orthologues to the human cell cycle genes from Tirosh et al. (2016) were used to score the cell cycle state of each cell using the scanpy.tl.score_genes_cell_cycle function. The list of catshark cell cycle genes is available in Table S2. Most cells were in G2/M or S phase of the cell cycle (93.9%), reflecting high proliferation rates (Fig. S13A). Notably, cells within the CNCC cluster were found in either S or G2/M phase, further supporting their extensive divisions as multipotent progenitors (Fig. S13B,C).

### Pseudotime

We searched for very early NCC marker genes within the early CNCC subcluster corresponding to specified non-delaminating, delaminating and migrating cells. These genes included *ZIC2, ZIC5, SOX10, FOXD3, PAX3* and *WNT1* (Soldatov et al., 2019). The rooting cell was randomly selected within the restricted group of cells that highly expresses the previously mentioned genes. Pseudotime was calculated for the CNCC subset (early CNCC, ectomesenchyme progenitors and ectomesenchyme) using the scanpy.tl.dpt function. Gene expression over pseudotime was plotted with matplotlib (matplotlib.org/stable/). Cells were arranged on the *x*-axis based on their pseudotime value, with the gene expression values plotted on the *y*-axis. A regression line was fit to the gene expression over pseudotime using the 'LinearGAM' function in pygam (pygam.readthedocs.io/en/latest/).

### Prediction of cellular communications

Predictions of cellular communications from scRNA-seq data were obtained using CellChat v2.1.0 (Jin et al., 2021). Catshark orthologues were mapped to their respective mouse genes to adapt the CellChat pipeline to the catshark scRNA-seq dataset. First, the mouse and catshark gff. and translated_cds fasta files from NCBI (www.ncbi.nlm.nih.gov/) were parsed using the 'parse_annotations' function from GENESPACE (Lovell et al., 2022) in order to produce a fasta file for each species containing the primary transcripts and headers with gene names. The obtained fasta files were used to acquire the best reciprocal hits between the mouse and catshark genomes using the 'diamond_protein_to_protein_best_reciprocal_hits' function from the rdiamond package (drostlab.github.io/rdiamond/index.html). In order to remove any one-to-many and many-to-many hits, only the top hit was kept. This resulted in 13,169 catshark orthologues (Table S3). The scRNA-seq dataset was subset to keep only the 13,169 orthologues identified, which were then renamed after their corresponding mouse genes. By doing this, we transferred the mouse gene name annotations to the catshark orthologues within the scRNA-seq dataset, which could then be used directly with the existing CellChat pipeline designed for mouse. CellChat was run on this adjusted catshark dataset following the standard vignette (Fig. S6). The netVisual_heatmap function was adapted to produce custom heatmaps displaying only significant signalling pathways of interest (*P*<0.05) (Fig. 3F).

### Immunofluorescence staining

Fixed embryonic samples were first bleached to remove autofluorescence by incubating them in Dent's Bleach solution [1 volume Vaprox (Steris, PB006EUR); 2 volumes Dent's Fix solution (80% methanol and 20% DMSO) (Roth, A994.1)] overnight on slow rotation. Afterwards, samples were washed three times for 10 min each in methanol and incubated overnight at 4°C in Dent's Fix solution. Embryos were then rinsed three times in 0.1% PBST and subsequently washed three times for 20 min each in 0.1% PBST to remove any traces of methanol and DMSO. After the washes, samples were incubated at room temperature for 5-7 days, depending on sample size, in rabbit anti-Sox9 primary antibody (Chemicon, Ab5535; 1:500) diluted in blocking solution composed of 20% DMSO and 5% donkey serum (Interchim, UP77719A-K) in PBS. Following primary antibody incubation, samples were rinsed three times in 0.1% PBST, washed three times for 20 min each in 0.1% PBST and incubated in secondary antibody at room temperature for 3 days diluted in blocking solution. The secondary antibody used was produced in donkey and conjugated with Alexa Fluor 594 (ThermoFisher, A21207; 1:1000). Nuclear staining was performed by incubating embryos overnight at 4°C in 1×DAPI (4′,6-diamidino-2-phenylindole; ThermoFisher, D21490) in PBST. Finally, samples were rinsed three times in 0.1% PBST, washed three times for 20 min each in 0.1% PBST and cleared using BABB (Becker et al., 2012). Whole-mount imaging was performed using glass-bottomed imaging dishes (Eppendorf, 0030740017) in a Zeiss LSM980 with Airyscan2 confocal microscope with the Plan-Apochromat 10×/0.45 M27 (Zeiss, 420640-9900) objective.

Following whole-mount imaging, stained embryos were removed from the imaging dishes, rinsed in methanol to remove the BABB solution and dehydrated overnight in 30% sucrose (Sigma S9378) in PBS on slow rotation. After dehydration, embryos were embedded in O.C.T. mounting medium (VWR, 361603E) in plastic moulds and stored at −20°C until sectioned. 30 µm cryosections were prepared using a Leica CM1860 cryostat and mounted on glass slides (Epredia J1830AMNZ). Cryosections were then

washed three times for 10 min each in PBST to remove the mounting media and overlaid with 87% glycerol (Labochem International, LC-7902.3) with a thin glass coverslip. Cryosections were imaged in a Zeiss LSM980 with Airyscan2 confocal microscope with the LD Plan-Neofluar 20×/0.4 M27 (Zeiss, 421350-9971-000) objective. Final confocal images were processed and exported from ZEN 3.9 software. To avoid fading of the fluorescence signal, embryos were sectioned and imaging carried out within 1 week.

### *In situ* HCR

*In situ* HCR in all embryonic samples was performed following the recommended protocol for whole-mount staining of the manufacturer (Molecular Instruments; HCRv3 FISH protocol) (Choi et al., 2018), with minor modifications. Briefly, fixed embryonic samples designated for whole-mount HCR staining were bleached to reduce autofluorescence, as described in the 'Immunofluorescence staining' section. Subsequently, samples were rehydrated in a decreasing methanol series for 10 min each (75%, 50%, 25% and 0% methanol in 0.1% PBST) at 4°C on slow rotation, post-fixed 15 min at room temperature in 4% PFA and pre-hybridized in 30% probe hybridization buffer for 30 min at 37°C. Once samples were equilibrated in probe hybridization buffer (sank to the bottom of the tube), they were incubated overnight at 37°C in 2 pmol of HCR probes diluted in 30% probe hybridization buffer in a thermomixer (Eppendorf, 5355) with gentle shaking (450 rpm). The following morning, samples were washed four times for 15 min each in 30% probe wash buffer at 37°C, four times for 15 min each in 0.1% 5×SSC-Tween (SSCT) at room temperature (20×SSC; Fisher Bioreagents, BP1325-4) and pre-amplified in HCR amplification buffer for 30 min. Once equilibrated, samples were incubated overnight at room temperature in 30 pmol of fluorescent-labelled hairpins diluted in amplification buffer, in the dark. On the following morning, samples were washed four times for 15 min each in 5×SSCT to remove excess hairpins and incubated overnight at 4°C in 1×DAPI in 5×SSCT to counterstain the nuclei. Finally, samples were washed four times for 15 min each in 5×SSCT, cleared using BABB and imaged using a Zeiss LSM980 with Airyscan2 confocal microscope.

Gene probe sets were either designed and purchased from Molecular Instruments or custom-designed using the HCR 3.0 probe maker (https://github.com/rwnull/insitu_probe_generator) and purchased as oligo pools from Integrated DNA Technologies. Fluorescent-labelled hairpins were purchased from Molecular Instruments and all buffers were prepared in-house following the manufacturer's protocol. Detailed information on the different gene probe sets used in this study can be found in Table S4.

### Synchrotron radiation micro-computed tomography (SRμCT)

Catshark embryos were contrasted in 1.5% phosphotungstic acid (PTA) solution (Sigma, P4006) diluted in 90% methanol following previously established protocols (Kaucka et al., 2017; Metscher, 2009). Embryos were stained for 1-3 weeks depending on their size: St.18-21 were stained for 1 week, St.23-28 for 2 weeks and St.29-31 for 3 weeks. Subsequently, embryos were washed twice in 90% ethanol overnight to remove any traces of the PTA contrasting solution and stored in 100% ethanol until scanning. Attenuation-contrast synchrotron radiation micro-computed tomography (SRμCT) measurements were acquired at the Imaging Beamline P05 of the storage ring PETRA III (Deutsches Elektronen Synchrotron – DESY, Hamburg, Germany) (Wilde et al., 2016). All embryos were imaged with a 20 keV photon energy and a sample-to-detector distance of 80 mm. St.18-25 embryos were imaged with a field of view (FOV) of 3.29 mm×2.47 mm and exposure time of 280 ms. St.27-31 embryos were imaged with a FOV of 6.57 mm×2.70 mm and exposure time of 80 ms. Tomographic reconstructions were performed using a custom pipeline implemented in MATLAB and the Astra Toolbox for the PETRA III Imaging Beamline P05 (Aarle et al., 2016; Moosmann et al., 2014). The resulting reconstructed images had an effective isotropic voxel size of 1.28 μm and 2.57 μm for each FOV, respectively. The CNCCs, facial ectomesenchyme, mesenchymal condensations and cartilage were manually segmented by a trained operator using Avizo3D Pro Software (ThermoFisher Scientific, Konrad-Zuse-Zentrum, Berlin, Germany). Distinction between mesenchymal condensations and cartilage in the SRμCT was performed following previously published information (Kaucka et al., 2017, 2018; Tesařová et al., 2016). Briefly, mesenchymal condensations appear as densely packed, homogenous regions with high absorption of the

PTA contrasting agent, while cartilage exhibits distinct cellular characteristics: round chondrocytes with reduced PTA absorption, lower cell density and a distinctive cellular layer surrounding the chondrocytes that displays high PTA absorption (Fig. S12).

### Genetic tracing of mouse cranial neural crest cells

*Sox10-creERT2/R26Confetti* mice were mated overnight and females were checked for the presence of a vaginal plug on the following morning. Upon reaching embryonic stage E8.5, genetic recombination was induced by intraperitoneal injection of pregnant females with tamoxifen (Sigma, T5648) dissolved in corn oil (Sigma C8267), which results in permanently fluorescent labelling of *Sox10*-expressing cells, primarily cranial neural crest cells. This strategy has been previously described (Kaucka et al., 2017). Tamoxifen concentration was 1 mg per animal. Embryos were then allowed to develop until reaching the desired embryonic stages (E9.5-E10.5). At this point, pregnant mice were sacrificed by cervical dislocation, and embryos were dissected in sterile ice-cold PBS under a stereomicroscope. Embryos were then fixed in 4% PFA for 1-3 h and incubated overnight at 4°C in DAPI solution to stain the nuclei. The following day, embryos were washed three times for 10 min each in PBS to remove excess DAPI and imaged using a Zeiss LSM980 with Airyscan2 confocal microscope. Fluorescently labelled embryos were first imaged laterally and then frontally to better observe the facial ectomesenchyme. Detailed information regarding the settings for the imaging of Confetti fluorescent proteins are described by Snippert et al. (2010).

### Small-spotted catshark gene nomenclature

The catshark genome assembly sScyCan1.1 (GCA_902713615.1) was used for all the analyses performed in this study. Despite being the most updated genome for this species and the reference assembly, it is still poorly curated and a large number of genes are annotated with a LOC number, which tremendously affects readability. We have manually curated the gene annotation and replaced the LOC numbers for symbols based on the genes' protein products. An 'l' was added at the end for those genes annotated as '-like'. For uncharacterized genes, BLASTp searches were performed against the mouse genome (*Mus musculus*, GRCm39 – GCF_000001635.27) and the hit with the lowest e-value and largest MAX score was selected as the most likely homologue. If BLASTp failed to detect a match, the gene was left with the LOC number annotation. A list of the gene symbols used throughout the text and their corresponding LOC annotation can be found in Table S5. Moreover, the sScyCan1.1 assembly follows the zebrafish gene nomenclature, which includes a letter at the end of the gene symbol to differentiate between paralogues derived from the teleost genome duplication event. Since the teleost genome duplication occurred after diverging from Chondrichthyes, many zebrafish paralogues are absent from the catshark genome and this nomenclature is misleading. Hence, we have adapted the gene symbols throughout the text to reflect those of non-teleost species.

### Statistics

Where statistical analysis was performed, details about the test (type of test, sample sizes and *P*-values) are indicated in the respective figure legends or the accompanying supplementary information.

### Acknowledgements

The authors thank Luiz Felipe Colli Sinhorini for technical support, animal caretakers Erika Teßmann and Ulrich Friese for their support with the catsharks in Ozeaneum (Stralsund), Martin Hansel and his team from Sea Life Berlin for providing catshark eggs, and the Max Planck Institute for Evolutionary Biology's Animal Facility for their support with the mice. The authors acknowledge Deutsches Elektronen-Synchrotron (DESY – Hamburg, Germany), a member of the Helmholtz Association HGF, for the provision of experimental facilities where SRμCT was carried out at PETRA III beamline P05 (proposal I-20230087). The authors thank Max Planck Society for the support.

### Competing interests

The authors declare no competing or financial interests.

### Author contributions

Conceptualization: M.K., E.E.-V.; Formal analysis: E.E.-V.; Funding acquisition: E.E.-V., M.K.; Investigation: M.K., E.E.-V., A.P.M.-R., L.W.G.S., A.-K.K., S.K.,

J.U.H., T.M.; Methodology: A.P.M.-R., L.W.G.S., J.U.H.; Resources: A.-K.K., C.F.-G., T.M.; Software: C.F.-G.; Supervision: M.K.; Visualization: E.E.-V.; Writing – original draft: M.K., E.E.-V.; Writing – review & editing: M.K., E.E.-V.

**Funding**
 Deposited in PMC for immediate release.

**Data and resource availability**
The single-cell RNA-sequencing dataset has been deposited at BioStudies (www.ebi.ac.uk/biostudies/) under accession number E-MTAB-15423. The single-cell transcriptomics dataset generated, together with all confocal microscopy images, are available in an interactive online resource at www.evolbio.mpg.de/escamilla-sharknc. The synchrotron radiation micro-computed tomography scans have been deposited in the Electron Microscopy Public Image Archive (EMPIAR) under accession number EMPIAR-12984). All other relevant data and details of resources can be found within the article and its supplementary information.

**Peer review history**
The peer review history is available online at https://journals.biologists.com/dev/lookup/doi/10.1242/dev.205258.reviewer-comments.pdf

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
