## [Peer Review File · Development (Cambridge, England)]

Developmental dynamics of catshark cranial neural crest cells provide insights into gnathostome facial evolution

Elio Escamilla-Vega, Andrea P. Murillo-Rincón, Louk W. G. Seton, Ann-Katrin Koch, Stella Kyomen, Carsten Fortmann-Grote, Jörg U. Hammel, Timo Moritz and Markéta Kaucká
DOI: 10.1242/dev.205258

Editor: Cassandra Extavour

Review timeline

Original submission: 19 September 2025
Editorial decision: 18 November 2025
Rebuttal submitted: 16 December 2025
Rebuttal decision: 5 January 2026
First revision: 20 February 2026
Accepted: 1 April 2026

Original submission

First decision letter

MS ID#: dev.205258

MS TITLE: Developmental dynamics of catshark cranial neural crest cells provide insights into gnathostome facial evolution

AUTHORS: Marketa Kaucka Petersen; Elio Escamilla-Vega; Andrea P. Murillo-Rincón; Louk W. G. Seton; Ann-Katrin Koch; Stella Kyomen; Carsten Fortmann-Grote; Jörg U. Hammel; Timo Moritz

Dear Dr Kaucka Petersen,

I have now received all the referees' reports on the above manuscript, and have reached a decision. I am sorry to say that the outcome is not a positive one. The referees' comments are appended below.

As you will see, the referees raise some significant concerns about your paper, and are not strongly in favour of publication. Having looked at the manuscript myself, I agree with their views, and I must therefore, reject your paper.

I do realise this is disappointing news, but Development receives many more papers than we can publish, and we can only accept manuscripts that receive strong support from referees.

I do hope you find the comments of the referees helpful, and that this decision will not dissuade you from considering Development for publication of your future work. Many thanks for sending your manuscript to Development.

Reviewer 1

SUMMARY OF THE ADVANCE MADE IN THIS PAPER AND ITS POTENTIAL SIGNIFICANCE TO THE FIELD

© 2026. Published by The Company of Biologists under the terms of the Creative Commons Attribution License (<https://creativecommons.org/licenses/by/4.0/>).

In this beautifully presented manuscript, the authors present single-cell transcriptomic data from the head region (otic level and above) of the catshark. They then focus on the cranial neural crest subset of cells. They validate expression of several genes by immunofluorescence and/or hybridization chain reaction in situ over developmental time. In addition, they use synchrotron tomography to examine anatomical changes. Their findings indicate that the cranial neural crest of the catshark is largely conserved compared to other gnathostomes but identify subtle lineage-specific adaptations. Overall, while this represents a good start and the paper is nicely done given the data in hand, it lacks novel insights and functional validation of novel candidate genes is missing. The following suggestions would improve the manuscript and make it a good candidate for publication in *Development*.

SUGGESTIONS TO AUTHORS

Major Revisions:

1. The authors dissected and dissociated the whole head region above the otic level (Fig. 2A), which encompasses a wide range of cell types of diverse embryonic origin, including placode-, mesoderm-, endoderm- and non-neural ectoderm-derived cells. As such, the "neural crest" cluster represents a very small fraction of the dataset. Although technically challenging, given that electroporation is feasible in the catshark (Fujimori et al., 2022), the authors should attempt to repeat their single-cell transcriptomic analysis using pure populations of cranial neural crest cells. These could be obtained by injection followed by electroporation of conserved enhancers (e.g., Sox10 or FoxD3 enhancers; e.g. Hockman et al., 2019 or Parker et al., 2019) that function cross species. This approach would provide a much deeper and more interpretable dataset.
2. The authors have not included the posterior cranial neural crest, which primarily populates the branchial arch region. This omission is problematic, as this head region represents an important component of the cranial neural crest. Therefore, the rationale for excluding it is unclear—particularly given that the title and scope of the study refer to the "cranial neural crest", rather than only the pre-otic subpopulations (mandibular and hyoid). Furthermore, in the dissections of later developmental stages (19 and 20), the hyoid arch derived from the pre-otic cranial neural crest appears to be missing. Is this indeed the case, or does the dashed line in Fig. 2A encompass this region?
3. Neural crest cells give rise to all of the glia of the peripheral nervous system. Why, then, is this neural crest derivative absent from the dataset? Although the authors dedicate a subsection to this topic, it warrants further validation given its biological importance, including the potential delayed emergence of this neural crest derivative. In its current form, the apparent absence of neuroglia further suggests that the cranial neural crest population is underrepresented in the current analysis.
4. While the authors focus on shared neural crest traits across gnathostomes, they also point to morphological differences. Are there candidate genes that reflect these differences? Highlighting and discussing such genes would strengthen the study and provide valuable insight into lineage-specific adaptations.
5. The study currently lacks functional testing of any genes identified by single-cell RNA-seq. Although I do understand that working with non-traditional model organisms represents significant technical challenges, and CRISPR-mediated loss-of-function is not yet established in sharks or skates, functional assays remain feasible. For instance, in vivo morpholino-mediated knockdowns could be employed to assess gene function, with resulting phenotypes analyzed using synchrotron tomography. Testing a few candidate genes in this manner would provide valuable insights whether gene functions are conserved or divergent across species, and would substantially strengthen the study, bringing it closer to the standards expected for publication in a journal such as *Development*.
6. The title does not reflect the content of the manuscript as the paper profiles part of the head and the neural crest component is a minor one.

Minor Revisions:

I recommend removing the reference to the "accelerated hyoid stream" in Fig. 7, as zebrafish do not possess an accelerated hyoid neural crest. If the authors wish to base their schematic on zebrafish cranial neural crest migration, this statement should either be removed or appropriately clarified in the text. The accelerated hyoid neural crest may instead represent a lineage-specific adaptation present only in certain fish lineages.

Although cartilaginous fishes occupy a key phylogenetic position as the closest living branch to the divergence between jawed and jawless vertebrates, the extant Chondrichthyes are themselves highly derived from their fossil ancestors. Therefore, I suggest tempering the statement in the Abstract: "little is known about CNCC properties in cartilaginous fishes (Chondrichthyes), leaving the ancestral condition of the first jawed vertebrates unresolved." The ancestral condition could instead be more appropriately inferred through comparative analyses among jawless vertebrates, Chondrichthyes, and the deepest extant branches of Osteichthyes.

Reviewer 2**SUMMARY OF THE ADVANCE MADE IN THIS PAPER AND ITS POTENTIAL SIGNIFICANCE TO THE FIELD**

This manuscript focuses on characterizing development of cranial neural crest cells in the chondrichthyan, *Scyliorhinus canicular* (small-spotted catshark), to evaluate the extent of conservation of these mechanisms in the sister group to Osteichthyes and uncover insights into the development of diverse craniofacial morphology among gnathostomes (jawed vertebrates). The question being posed is interesting, as several papers have focused related questions on other jawed species including bony fish, but much less is known about neural crest cells in cartilaginous fishes. The researchers used molecular, genetic, and morphological approaches to substantiate the conservation of core developmental gene programs for early CNCC development in their model organism. They also show that diverse craniofacial morphology in Chondrichthyes is likely underscored by divergent behavior of ectomesenchymal progenitors during early embryogenesis. This research will provide valuable information to the fields of neural crest cell biology and EvoDevo as it illuminates potential novel mechanisms driving craniofacial development. It is also a valuable resource for studying early embryogenesis at the molecular level in a non-traditional model organism.

SUGGESTIONS TO AUTHORS

Major comments: n/a

Minor comments:

1. It is not clear why the top row in figure 1 is included because the higher mag or close up images in the figure (G) are such better resolution. Row A seems like it could be placed in supplemental without much loss to the manuscript.
2. Line 17 says that primary neurulation occurs in "primary neurulation, a process observed in amniotes, bichirs, and sterlets," but it is not clear why they limit this to these species? Several anamniotes including amphibians (frogs and axolotls) and zebrafish (Werner et al., 2021) use primary neurulation as well.
3. In the introduction, there is mention of "shark-specific facial morphology" and in the section of the results titled "The periocular ectomesenchyme of the catshark is homologous to the frontonasal ectomesenchyme in Osteichthyes" a similar phrase is used: "characteristic Chondrichthyes facial morphology." These statements are in direct conflict with a claim made in the discussion section stating "Chondrichthyes display an impressive variety of facial morphologies..." The authors should clarify why the small-spotted catshark is a representative species of the chondrichthyans that is beneficial to study, while also addressing the limitations associated with drawing conclusions about an entire phylogenetic group based on a single species.
4. An overall observation for the main figures: arrowheads, asterisks, or some other annotation should be added in the main figures to highlight key characteristics of the images

discussed by the authors as it is sometimes unclear what feature in the images is being discussed in the text. This could significantly improve clarity of the results.

5. There seems to be a discrepancy between the data for MDGA2 in the dot plot in Figure 3C and the expression pattern of MDGA2 in Fig. 3D- specifically- in the dot plot it appears to be groups with early CNCC but in the expression image- it appears to be late and distinct from the UNC5C expression which mirrors Sox9 from figure 1. This is an example of where arrowheads or asterisks could be useful to clarify the relevant regions of expression (or absence of expression) that are discussed in the text. Sections should also be included if they show the co-localization better.

6. Figure 5 could benefit from more annotations as well, but the images are beautiful.

7. There are several sentences at the end of the second paragraph of the results section titled "CNCC migratory streams are conserved in the catshark" that are repetitive with earlier information in that same paragraph. These sentences, which focus on the hyoid and branchial CNCC migratory streams, could be removed or condensed to improve clarity.

8. "To validate cluster identities, we performed whole-mount HCR in situ hybridization using St. 19 embryos as a representative stage for the dataset." What is the justification for using a single stage here rather than several stages? Is this sufficient to validate the complex expression landscape captured in your scRNAseq experiments?

9. In the "CNCC-derived neuroglia are not detected in the scRNAseq analysis" section of the results, there is a grammatical error in the sentence that states "To investigate this cell lineage, we first examined the expression of PHOX2B, a canonical marker of autonomic neuronal (Fig. S8A)." This sentence should say autonomic neurons.

10. In the same paragraph, there is a sentence stating, "In mice, late-migratory CNCC give rise to neuroglia of the cranial ganglia." It should be clarified that differentiation of the CNCC into neuroglial derivatives occurs sequentially after ectomesenchymal fate specification and if this is a conserved trait in non-mammalian species as well.

11. "Nonetheless, the robust ALX1 expression in the catshark periocular mesenchyme supports its transcriptional equivalence to the frontonasal mesenchyme found in other Osteichthyes." This statement is a bit overreaching. Demonstrating functional equivalence between these two populations would provide the necessary evidence to support this claim.

12. "EDNRB is expressed in early CNCC and becomes restricted to the PNS by St. 21." Was this statement confirmed via coexpression of EDNRB with a marker of PNS cell types or is this conclusion purely observational?

13. "The conserved expression patterns of the endothelin receptors in the catshark suggest functional conservation of this signaling pathway across vertebrates." This statement as written is overreaching. It should be clarified that this conclusion cannot be made without experiments confirming the function of each receptor.

14. In Figure 6, it is unclear how the classification of "mesenchymal condensations" versus "cartilage" is assigned to the images. Please clarify.

Rebuttal submitted

Dear Prof. Dr. Extavour, dear *Development* Editorial Office,

We cordially thank you for considering our manuscript MS# dev.205258 (Developmental dynamics of catshark cranial neural crest cells provide insights into gnathostome facial evolution) and for the time invested by the reviewers in providing thoughtful and detailed evaluations. We appreciate the generally positive assessment of our work, particularly the recognition that our study provides a valuable molecular and morphological resource for cranial neural crest cell (CNCC) development in a key non-model vertebrate, the small spotted catshark.

We respectfully submit this letter to request reconsideration of our manuscript in light of the detailed responses and revisions we have provided. Below, we summarize the main issues raised and explain why we believe the manuscript meets the standards for publication in *Development*.

1. Requests for experimental approaches that are currently infeasible in chondrichthyans

Several major reviewer requests (e.g. CNCC-specific scRNA-seq via enhancer-driven electroporation, gene-specific functional assays using morpholinos or CRISPR/Cas9 gene editing strategy) would undoubtedly be valuable if technically possible. However, as we document in detail in our responses, these approaches are not currently feasible in chondrichthyan embryos, particularly at the early developmental stages analyzed in this study.

To our knowledge, electroporation in chondrichthyans has only been achieved in late-stage embryos or adults, and no published work has demonstrated enhancer-driven labeling, morpholino knockdown, or gene-specific functional perturbation *in vivo* at early embryonic stages in sharks or skates. Establishing such methods would require substantial and independent methodological development, well beyond the scope of the present study and beyond the current technical capabilities of the field (1-4).

We emphasize that these limitations are not specific to our laboratory, but rather reflect well-recognized constraints inherent to working with early chondrichthyan embryos, including limited embryo availability, extreme sensitivity to manipulation during early developmental stages, and the absence of established genetic tools (2, 4-7). We have clarified these points explicitly in the revised manuscript and the Response to reviewers.

2. Scope and biological rationale of the single-cell dataset

The reviewers raised concerns regarding (i) the whole-head dissociation strategy and (ii) the exclusion of posterior (branchial) CNCC from the single-cell dataset. These choices were deliberate and biologically motivated, not omissions.

Our whole-head strategy enabled us to infer CNCC interactions with surrounding tissues, including mesoderm, CNS, and ectoderm, allowing us, for example, to infer cellular communication at this crucial developmental event, where multi-tissue coordination is key to the establishment of key body structures. This would not have been possible in a CNCC-only dataset. Moreover, the number of CNCC recovered is comparable to that in CNCC-focused single-cell studies of established model organisms at equivalent stages (8-10), and the dataset robustly captures early CNCC states and ectomesenchymal trajectories. Generally, the recovered cell populations and trajectories are fully aligned with results from other vertebrate models (8, 9, 11), demonstrating the robustness and value of the presented dataset.

The restriction of single-cell analysis to the pre-otic (trigeminal) CNCC stream reflects the study's focus on facial development and evolution. However, please note that imaging-based analyses (HCR, immunofluorescence) encompass all CNCC streams and an extended developmental timeline, thereby preserving the full developmental context. We have clarified this rationale and terminology throughout the revised manuscript.

3. Absence of CNCC-derived neuroglia in the scRNA-seq dataset

The apparent absence of CNCC-derived neuroglia was raised as a concern; however, this reflects the developmental timing, rather than an underrepresentation of CNCC. Our single-cell sampling window (St. 16-20) precedes the emergence of peripheral neuroglia in chondrichthyans, which we independently demonstrate to occur at St. 21-22 using *PHOX2B* HCR. This timing is consistent with known temporal hierarchies of CNCC differentiation across vertebrates and mirrors observations in other developmental single-cell atlases (8, 9, 11).

4. Functional validation and interpretation of conservation

While the reviewers requested gene-level functional validation, we have carefully adjusted the manuscript to avoid overinterpretation. Claims of “functional equivalence” or “functional conservation” have been appropriately tempered, and we now clearly distinguish between transcriptional, spatial, and morphological correspondence versus direct functional evidence. Where relevant, we highlight candidate pathways and genes (e.g., identified via CellChat

analyses) as hypotheses for future functional testing rather than definitive conclusions.

5. Overall contribution and suitability for *Development*

We believe the revised manuscript provides a substantive and timely contribution to the fields of neural crest biology and evolutionary developmental biology by:

- Delivering the first early-stage single-cell analyses in a cartilaginous fish;
- Integrating scRNA-seq with high-resolution spatial validation (HCR, immunofluorescence) and synchrotron-based 3D reconstructions;
- Demonstrating strong conservation of early CNCC programs across gnathostomes while identifying lineage-specific developmental features relevant to facial evolution;
- Establishing a foundational framework upon which future functional and comparative studies in chondrichthyans can build;
- Providing open-source scRNAseq data and 3D gene expression mapping to the research community in a user-friendly interactive database to boost research in chondrichthyans.

We respectfully suggest that requiring experimental methodologies that are not yet achievable in this lineage would set a standard that effectively precludes mechanistic developmental studies in non-traditional vertebrate models. We believe *Development* has historically played a leading role in supporting rigorous, conceptually driven work in such systems when technical constraints are clearly articulated and conclusions are appropriately framed.

For these reasons, we kindly ask you to reconsider our manuscript for publication in *Development*. We would, of course, be happy to address any remaining concerns you may have.

We attach a draft of our Response to the Reviewers, and we would be prepared to submit all revised material within a few weeks.

Thank you very much for your time and consideration.

With many thanks & best wishes,

Markéta

References:

1. C. Fujimori, *et al.*, In vitro and in vivo gene transfer in the cloudy catshark *Scyliorhinus torazame*. *Development, Growth & Differentiation* **64**, 558-565 (2022).
2. J. A. Gillis, *et al.*, “Big insight from the little skate: *Leucoraja erinacea* as a developmental model system” in *Current Topics in Developmental Biology*, Emerging Model Systems in Developmental Biology., B. Goldstein, M. Srivastava, Eds. (Academic Press, 2022), pp. 595-630.
3. H. Jung, *et al.*, The Ancient Origins of Neural Substrates for Land Walking. *Cell* **172**, 667-682.e15 (2018).
4. Y. Lund-Ricard, A. Boutet, “Current Trends in Chondrichthyes Experimental Biology” in *Handbook of Marine Model Organisms in Experimental Biology*, (CRC Press, 2021).
5. L. Adams, *et al.*, Monitoring egg fertility, embryonic morbidity, and mortality in an oviparous elasmobranch using ultrasonography. *Front. Vet. Sci.* **11** (2024).
6. W. W. Ballard, J. Mellinger, H. Lechenault, A series of normal stages for development of *Scyliorhinus canicula*, the lesser spotted dogfish (*Chondrichthyes: Scyliorhinidae*). *J Exp Zool* **267**, 318-336 (1993).

7. K. Onimaru, F. Motone, I. Kiyatake, K. Nishida, S. Kuraku, A staging table for the embryonic development of the brownbanded bamboo shark (*Chiloscyllium punctatum*). *Developmental Dynamics* **247**, 712-723 (2018).
8. R. Soldatov, *et al.*, Spatiotemporal structure of cell fate decisions in murine neural crest. *Science* **364**, eaas9536 (2019).
9. S. Menchero, *et al.*, Marsupial single-cell transcriptomics identifies temporal diversity in mammalian developmental programs. *Developmental Cell* **60**, 3339-3356.e5 (2025).
10. G. La Manno, *et al.*, Molecular architecture of the developing mouse brain. *Nature* **596**, 92-96 (2021).
11. D. Tatarakis, *et al.*, Single-cell transcriptomic analysis of zebrafish cranial neural crest reveals spatiotemporal regulation of lineage decisions during development. *Cell Rep* **37**, 110140 (2021).

Response to the Reviewers

We thank the Reviewers for their careful and thoughtful evaluation of our manuscript and for their constructive suggestions. We have revised the manuscript accordingly and respond to each comment below (in blue).

Reviewer #1

In this beautifully presented manuscript, the authors present single-cell transcriptomic data from the head region (otic level and above) of the catshark. They then focus on the cranial neural crest subset of cells. They validate expression of several genes by immunofluorescence and/or hybridization chain reaction in situ over developmental time. In addition, they use synchrotron tomography to examine anatomical changes. Their findings indicate that the cranial neural crest of the catshark is largely conserved compared to other gnathostomes but identify subtle lineage-specific adaptations. Overall, while this represents a good start and the paper is nicely done given the data in hand, it lacks novel insights and functional validation of novel candidate genes is missing. The following suggestions would improve the manuscript and make it a good candidate for publication in *Development*.

We would like to cordially thank Reviewer #1 for their positive assessment of the manuscript and for their valuable comments and suggestions, which helped us improve the clarity, limitations, and interpretation of the findings.

Major comments:

1. The authors dissected and dissociated the whole head region above the otic level (Fig. 2A), which encompasses a wide range of cell types of diverse embryonic origin, including placode-, mesoderm-, endoderm- and non-neural ectoderm-derived cells. As such, the "neural crest" cluster represents a very small fraction of the dataset. Although technically challenging, given that electroporation is feasible in the catshark (Fujimori *et al.*, 2022), the authors should attempt to repeat their single-cell transcriptomic analysis using pure populations of cranial neural crest cells. These could be obtained by injection followed by electroporation of conserved enhancers (e.g., *Sox10* or *FoxD3* enhancers; e.g. Hockman *et al.*, 2019 or Parker *et al.*, 2019) that function cross species. This approach would provide a much deeper and more interpretable dataset.

We appreciate the reviewer's thoughtful suggestion to enrich cranial neural crest cells (CNCC) using electroporation-based genetic labeling. We agree that such approach would be extremely valuable if technically feasible. However, we would like to clarify that this methodology is not currently applicable to early-stage chondrichthyan embryos, particularly at the stages when CNCC specification, delamination, and early migration occur.

To date, electroporation in chondrichthyans has only been reported in adult animals or late-stage embryos (e.g., St. 31-32 in the cloudy catshark; Fujimori et al., 2022) (1), and more recently, in St. 27-28 little skate embryos (2). To our knowledge, these reports represent the entirety of the published literature on electroporation in this phylogenetic lineage *in vivo*, and no study has achieved electroporation at early embryonic stages comparable to those analyzed in our study (St. 16-20).

Early chondrichthyan embryos differ fundamentally from later stages: they cannot be cultured outside the egg case, are extremely sensitive to environmental perturbations, and do not survive the manipulations associated with electroporation (3-6). Even minimal disturbance of the yolk sac is typically lethal at these early stages. By contrast, embryos from St.26 onward are substantially more resilient, consistent with the stages used in the published studies cited above (3, 5, 6). In a recent book chapter on non-model organisms by leaders in the field, it is explicitly noted that experimental manipulation of early chondrichthyan embryos remains under development and lacks robust, reproducible protocols (5).

In addition, the enhancers suggested by the reviewer (e.g., Sox10, FoxD3) have never been functionally validated *in vivo* in chondrichthyans, precisely because early-stage genetic manipulation is not yet achievable. Establishing enhancer-driven labeling at these stages would therefore require extensive method development and validation, constituting an independent research program well beyond the scope of the present study.

From a biological perspective, we also note that enriching exclusively for CNCC would eliminate important contextual information. Our whole-head strategy enabled us to analyze interactions between CNCC and their surrounding tissues, including mesoderm, CNS, and ectoderm, as demonstrated by the presented CellChat analysis (Fig. 3G). These inter-tissue signaling relationships would not be accessible in a CNCC- only dataset. Importantly, the proportion of CNCC recovered reflects their *in vivo* abundance at these stages.

This approach is consistent with recent single-cell studies in other systems. For example, whole-embryo profiling of marsupial CNCC recovered early, migratory and ectomesenchymal populations without sensory or autonomic derivatives at comparable stages, in agreement with our findings (7). Even genetically enriched CNCC datasets, such as Soldatov et al. (Science, 2019) (8), recovered similar numbers of CNCC at equivalent developmental stages (approximately 500 at mouse E8.5 versus 353 in our dataset). We further complement our single-cell data with extensive spatial validation (HCR, immunofluorescence) and synchrotron-based 3D reconstructions spanning later developmental stages, providing a more comprehensive picture of developmental dynamics.

Finally, we emphasize the severe practical constraints of working with chondrichthyan embryos. Catsharks produce very limited numbers of embryos, with strong seasonal variation and high embryonic mortality and morbidity (3, 5, 6, 9). These constraints make repetition or expansion of early-stage single-cell experiments exceptionally challenging and explain why such datasets remain exceedingly rare in this lineage. To the best of our knowledge, this is the first early developmental dataset from any chondrichthyan species.

Taken together, while we fully agree with the conceptual value of the reviewer's suggestion and we would be generally extremely enthusiastic about the opportunity to perform the first-ever (successful) electroporation and genetic tracing in early embryos of the small spotted catshark, the proposed experiment is currently not feasible in chondrichthyans. We respectfully maintain that our whole-head single-cell strategy provides a robust, biologically meaningful, and unique dataset that is well-aligned with the current technical state of the field.

2. The authors have not included the posterior cranial neural crest, which primarily populates the branchial arch region. This omission is problematic, as this head region represents an important component of the cranial neural crest. Therefore, the rationale for excluding it is unclear—particularly given that the title and scope of the study refer to the "cranial neural crest", rather than only the pre-otic subpopulations (mandibular and hyoid). Furthermore, in the dissections of later developmental stages (19 and 20), the hyoid arch derived from the pre-otic cranial neural crest appears to be missing. Is this indeed the case, or does the dashed line in Fig. 2A encompass this region?

We thank the reviewer for highlighting the diversity of CNCC populations. As correctly noted, CNCC are subdivided into trigeminal (mandibular), hyoid, and branchial streams, each contributing to distinct cranial structures. Trigeminal CNCC generate the first pharyngeal arch, facial skeleton, and anterior cranial ganglia; hyoid CNCC contribute to second arch derivatives and components of the middle ear; and branchial CNCC populate arches 3-7 in the catshark, forming structures such as the gill cartilage and posterior cranial ganglia (10-12). While branchial CNCC indeed represent an essential component of the head, they do not contribute to the development of facial structures.

Our study specifically focuses on facial evolution, which is generated exclusively by the trigeminal CNCC stream. For this reason, our single-cell analysis and synchrotron-based reconstructions were deliberately restricted to the pre-otic region. This is a conceptual decision aligned with the biological question under investigation, rather than an omission.

We respectfully suggest that changing the title from “cranial neural crest” to “pre-otic neural crest” would introduce unnecessary confusion for the readers; the term “cranial neural crest” is conventionally used to describe all anterior NCC streams (8, 10, 11), and our revised manuscript explicitly explains why the trigeminal CNCC are the focus of the study and the core of the evolutionary question.

Importantly, the study does not exclude hyoid or branchial CNCC from morphological or molecular analyses. All imaging-based datasets (immunofluorescence and HCR) encompass the entire embryonic head, including all CNCC streams, and our descriptions of delamination, early migration (Fig. 1), and gene expression (Fig. 3D; Fig. 4) include the trigeminal, hyoid, and branchial populations.

Regarding the reviewer’s concern about potential inclusion of hyoid CNCC in the single-cell dataset at St. 19-20, we screened for *HOX* gene expression, which reliably marks hyoid and branchial CNCC in the catshark (13). This analysis, now included as a new Supplementary Figure (Fig. S2), show that *HOX* expression is confined to hindbrain-associated clusters and absent from the CNCC cluster, confirming that the dataset contains only trigeminal CNCC. The dashed boundaries in Fig. 2A therefore accurately reflect the dissected region, and the hyoid population is not represented in the single-cell dataset.

In summary, the exclusion of posterior CNCC from the single-cell dataset was a deliberate decision to align with the scope of the study. The morphological and gene expression analyses encompass all CNCC streams, ensuring that the developmental context of the trigeminal population is adequately represented. We have clarified this rationale more explicitly in the revised manuscript to avoid confusion.

3. Neural crest cells give rise to all of the glia of the peripheral nervous system. Why, then, is this neural crest derivative absent from the dataset? Although the authors dedicate a subsection to this topic, it warrants further validation given its biological importance, including the potential delayed emergence of this neural crest derivative. In its current form, the apparent absence of neuroglia further suggests that the cranial neural crest population is underrepresented in the current analysis.

We agree that CNCC-derived neuroglia represent an important lineage. Their absence from the single-cell dataset reflects developmental timing, not underrepresentation of CNCC.

The presented scRNA-seq dataset spans St. 16-20, a window during which CNCC undergo delamination, migration, and early differentiation toward ectomesenchymal fates. In vertebrates, neuroglial derivatives arise later than ectomesenchymal lineages, a temporal hierarchy that has been documented in multiple species (7, 8, 14).

Consistent with this, we performed *PHOX2B* HCR analysis and found that the first CNCC-derived neuroglia appear at St. 21-22, outside the window of our single-cell sampling. Thus, their absence from the dataset is expected and biologically informative rather than indicative of technical bias. We note that similar stage-dependent absences are common in developmental single-cell studies, including recent studies, such as marsupial CNCC atlas (7). Similarly to our work, the marsupial NCC dataset did not recover sensory-lineage derivatives, not because opossum CNCC fail to produce them, but because the

latest stage analyzed (E10.5) precedes their emergence (7). Our dataset reflects the same principle.

The embryonic stages we selected for single-cell profiling were guided by SOX9 immunofluorescence and morphological comparisons with other vertebrates; however, the emergence of neuroglia at St.21-22 could not have been predicted beforehand. Such discoveries are a natural (and valuable) outcome of exploratory developmental studies and point to exciting avenues for future research, as acknowledged in our Discussion. Importantly, the timing of neuroglial emergence in chondrichthyans has not been previously documented. Our study, therefore, provides the first empirical evidence that places this transition at St. 21-22.

We have clarified this point and expanded the discussion of developmental timing (see lines 658-662) and the chapter “limitations” in the revised manuscript.

4. While the authors focus on shared neural crest traits across gnathostomes, they also point to morphological differences. Are there candidate genes that reflect these differences? Highlighting and discussing such genes would strengthen the study and provide valuable insight into lineage-specific adaptations.

Thank you for this interesting question and suggestion. If the opportunity to resubmit this work arises, we will perform additional bioinformatical comparative analyses comparing the molecular programs of CNCC in small-spotted catshark and other model organisms (e.g., zebrafish and mouse) using publicly available data. We will identify the components unique (or absent) in the small-spotted catshark CNCC and perform thorough validation using HCR, aiming to identify candidate genes underlying lineage-specific features. We will then expand the Results and Discussion to highlight candidate signaling pathways that may be associated with lineage-specific adaptations. By performing preliminary analysis, we noticed a differential involvement of periostin- and TULP-associated signaling in catshark ectomesenchymal populations. We will validate these observations and discuss these particular pathways explicitly as candidate contributors to species-specific craniofacial patterning, while clearly framing them as hypotheses for future functional investigation.

5. The study currently lacks functional testing of any genes identified by single-cell RNA-seq. Although I do understand that working with non-traditional model organisms represents significant technical challenges, and CRISPR-mediated loss-of-function is not yet established in sharks or skates, functional assays remain feasible. For instance, *in vivo* morpholino-mediated knockdowns could be employed to assess gene function, with resulting phenotypes analyzed using synchrotron tomography. Testing a few candidate genes in this manner would provide valuable insights whether gene functions are conserved or divergent across species, and would substantially strengthen the study, bringing it closer to the standards expected for publication in a journal such as Development.

We fully agree that functional assays would greatly advance understanding of gene function in chondrichthyan CNCC development. However, gene-specific functional perturbation has not yet been achieved in chondrichthyan embryos, including morpholino-mediated knockdown.

To our knowledge, no published study has demonstrated successful morpholino delivery or efficacy in sharks or skates, and the same biological and technical constraints that limit CRISPR approaches also apply to morpholinos. Early embryos are not amenable to one-cell injection, genetic manipulation, or prolonged *ex vivo* culture. As such, the suggested experiments are not currently feasible.

At present, functional approaches in early chondrichthyan embryos are mostly limited to fate mapping using lipophilic dyes and broad pharmacological perturbations (5), which do not permit gene-specific or tissue-specific loss-of-function analyses. We have clarified these constraints in the Discussion.

We certainly do share the reviewer’s enthusiasm for future methodological advances in this area, but emphasize that the absence of functional assays reflects field-wide technical limitations, not an omission specific to this study. We cordially thank the Reviewer for their understanding!

6. The title does not reflect the content of the manuscript as the paper profiles part of the head and

the neural crest component is a minor one.

We respectfully disagree. The study is explicitly centered on CNCC and their role in facial evolution, as stated in the title. While CNCC constitute a subset of the single-cell dataset, they do form the conceptual and biological focus of the manuscript. The study integrates scRNA-seq with extensive spatial and morphological analyses, focusing on CNCC specification, migration, signaling interactions, and their contribution to facial morphogenesis.

Importantly, the manuscript presents a comprehensive analysis of CNCC beyond their representation in the single-cell dataset. We provide detailed descriptions of CNCC' specification, delamination, and migratory trajectories, as well as inference of signaling interactions between CNCC and surrounding cranial tissues. In addition, the study includes extensive gene expression mapping that encompass all CNCC subpopulations, not solely the trigeminal stream highlighted in the single-cell experiment, and synchrotron-based 3D reconstructions that illuminate facial morphogenesis.

Thus, although the single-cell component necessarily concentrates on trigeminal CNCC, the broader manuscript examines CNCC biology at multiple levels and across multiple methodologies. For these reasons, we believe that the current title accurately reflects the scope and emphasis of the work.

Minor comments:

All minor comments from Reviewer #1 have been addressed as suggested. Figures have been updated, text has been clarified or toned down where appropriate, and overstatements have been corrected. Specific changes are detailed in the revised manuscript.

1. I recommend removing the reference to the "accelerated hyoid stream" in Fig. 7, as zebrafish do not possess an accelerated hyoid neural crest. If the authors wish to base their schematic on zebrafish cranial neural crest migration, this statement should either be removed or appropriately clarified in the text. The accelerated hyoid neural crest may instead represent a lineage-specific adaptation present only in certain fish lineages.

In response to this comment, we have modified the figure and clarified the information in the main text and the corresponding figure legend. Please see the adapted Figure 7 and its respective legends, and see the lines 531-537.

2. Although cartilaginous fishes occupy a key phylogenetic position as the closest living branch to the divergence between jawed and jawless vertebrates, the extant Chondrichthyes are themselves highly derived from their fossil ancestors. Therefore, I suggest tempering the statement in the Abstract: "little is known about CNCC properties in cartilaginous fishes (Chondrichthyes), leaving the ancestral condition of the first jawed vertebrates unresolved." The ancestral condition could instead be more appropriately inferred through comparative analyses among jawless vertebrates, Chondrichthyes, and the deepest extant branches of Osteichthyes.

We agree with your comment and have thus removed this statement from the abstract.

Closing remark:

We believe that the revised manuscript addresses the reviewers' concerns while clearly articulating the current technical boundaries of developmental research in chondrichthyans. Importantly, the study leverages the most advanced and reliable methodologies currently available for early-stage shark embryos, combining single-cell transcriptomics with high-resolution spatial validation and synchrotron-based 3D reconstruction to extract maximal biological insight from a system where experimental manipulation remains extremely limited.

Historically, the journal Development has played a central role in advancing rigorous, conceptually driven developmental biology in non-traditional model organisms by carefully valuing executed descriptive and comparative studies, provided technical constraints are transparently acknowledged,

and conclusions are appropriately framed. We believe our work aligns well with this tradition, as it provides a foundational molecular and morphological framework for cranial neural crest development in chondrichthyans, motivates future methodological and functional studies, and establishes reference data that will be essential once new experimental tools become available.

For these reasons, we respectfully resubmit with the conviction that the manuscript meets the rigorous standards of transparency and conceptual contribution expected for publication in Development.

Reviewer #2

We thank Reviewer #2 for the positive and encouraging assessment of our study and for the constructive suggestions.

All minor comments from Reviewer #2 have been addressed. Figures have been annotated for clarity, text has been revised to improve precision and consistency, and limitations regarding species representation and functional inference have been explicitly acknowledged in the Discussion.

This manuscript focuses on characterizing development of cranial neural crest cells in the chondrichthyan, *Scyliorhinus canicula* (small-spotted catshark), to evaluate the extent of conservation of these mechanisms in the sister group to Osteichthyes and uncover insights into the development of diverse craniofacial morphology among gnathostomes (jawed vertebrates). The question being posed is interesting, as several papers have focused related questions on other jawed species including bony fish, but much less is known about neural crest cells in cartilaginous fishes. The researchers used molecular, genetic, and morphological approaches to substantiate the conservation of core developmental gene programs for early CNCC development in their model organism. They also show that diverse craniofacial morphology in Chondrichthyes is likely underscored by divergent behavior of ectomesenchymal progenitors during early embryogenesis. This research will provide valuable information to the fields of neural crest cell biology and EvoDevo as it illuminates potential novel mechanisms driving craniofacial development. It is also a valuable resource for studying early embryogenesis at the molecular level in a non-traditional model organism.

Minor comments:

1. It is not clear why the top row in figure 1 is included because the higher mag or close up images in the figure (G) are such better resolution. Row A seems like it could be placed in supplemental without much loss to the manuscript.

We thank the reviewer for this suggestion. We included the top row of Figure 1 (panel A) to provide a developmental and anatomical overview across multiple intermediate stages that cannot be easily conveyed by high-magnification images alone. These lower-magnification views allow readers to appreciate the spatial relationship between cranial and trunk NCC populations and to contextualize the higher-resolution panels.

In addition, Figure 1A serves an important organizational purpose: it contains the dashed reference lines indicating the positions of panels B-F", while 1G labels key anatomical structures. Including all information within a single high-magnification panel would substantially reduce clarity. For these reasons, we believe that retaining panel A in the main figure improves readability and accessibility, particularly for readers less familiar with catshark embryonic anatomy.

2. Line 17 says that primary neurulation occurs in "primary neurulation, a process observed in amniotes, bichirs, and sterlets," but it is not clear why they limit this to these species? Several anamniotes including amphibians (frogs and axolotls) and zebrafish (Werner et al., 2021) use primary neurulation as well.

We thank the reviewer for pointing out this imprecision. We agree that the original wording was unnecessarily restrictive and inadvertently excluded several anamniote species in which primary neurulation also occurs, including amphibians and zebrafish. We have revised the

sentence to more accurately reflect the broader distribution of primary neurulation across vertebrates.

3. In the introduction, there is mention of "shark-specific facial morphology" and in the section of the results titled "The periocular ectomesenchyme of the catshark is homologous to the frontonasal ectomesenchyme in Osteichthyes" a similar phrase is used: "characteristic Chondrichthyes facial morphology." These statements are in direct conflict with a claim made in the discussion section stating "Chondrichthyes display an impressive variety of facial morphologies..." The authors should clarify why the small-spotted catshark is a representative species of the chondrichthyans that is beneficial to study, while also addressing the limitations associated with drawing conclusions about an entire phylogenetic group based on a single species.

We appreciate this thoughtful observation and the opportunity to clarify our intent. Our references to "shark-specific" or "characteristic chondrichthyan" facial morphology were meant to highlight adult morphological outcomes that ultimately derive from CNCC, thereby emphasizing the evolutionary relevance of studying facial ectomesenchyme.

By contrast, our discussion of the impressive diversity of chondrichthyan facial morphologies refers explicitly to adult forms across the clade, which indeed show substantial variation. These statements are therefore not contradictory but refer to morphology at different developmental stages. As in other vertebrate groups, embryos within a lineage are far more similar to one another during early development than their adult anatomies might suggest.

At the embryonic stages analyzed here, available data from multiple chondrichthyan species (including small-spotted catshark, cloudy catshark, little skate, and elephant shark) indicate a high degree of conservation in the organization of periocular and facial ectomesenchyme (12, 15, 16). This supports the use of the small-spotted catshark as a representative and experimentally accessible model for early chondrichthyan development, while fully acknowledging that no single species can capture the full morphological diversity of an entire clade.

We have expanded the Discussion to clarify this distinction and to explicitly acknowledge the limitations associated with drawing broader evolutionary inferences from a single species.

4. An overall observation for the main figures: arrowheads, asterisks, or some other annotation should be added in the main figures to highlight key characteristics of the images discussed by the authors as it is sometimes unclear what feature in the images is being discussed in the text. This could significantly improve clarity of the results.

We fully agree and thank the reviewer for this helpful suggestion. We have added annotations throughout the main and supplementary figures to more clearly highlight the features discussed in the text and improve figure readability.

5. There seems to be a discrepancy between the data for MDGA2 in the dot plot in Figure 3C and the expression pattern of MDGA2 in Fig. 3D- specifically- in the dot plot it appears to be groups with early CNCC but in the expression image- it appears to be late and distinct from the UNC5C expression which mirrors Sox9 from figure 1. This is an example of where arrowheads or asterisks could be useful to clarify the relevant regions of expression (or absence of expression) that are discussed in the text. Sections should also be included if they show the co-localization better.

We thank the reviewer for drawing attention to this point and agree that the original presentation could be confusing. The "early CNCC" cluster encompasses a continuum of CNCC states, including non- delaminating, delaminating, and migrating cells, which are not resolved into distinct subclusters at the chosen resolution.

As a result, although both *UNC5C* and *MDGA2* mark this cluster, they are expressed in different transitional CNCC subpopulations. To avoid misinterpretation, we have removed this panel and revised the text accordingly.

6. Figure 5 could benefit from more annotations as well, but the images are beautiful.

We thank the reviewer for the positive feedback and have added additional annotations to Figure 5 to improve clarity.

7. There are several sentences at the end of the second paragraph of the results section titled "CNCC migratory streams are conserved in the catshark" that are repetitive with earlier information in that same paragraph. These sentences, which focus on the hyoid and branchial CNCC migratory streams, could be removed or condensed to improve clarity.

We agree with the reviewer and have condensed this paragraph by removing repetitive sentences, improving clarity and flow.

8. "To validate cluster identities, we performed whole-mount HCR in situ hybridization using St. 19 embryos as a representative stage for the dataset." What is the justification for using a single stage here rather than several stages? Is this sufficient to validate the complex expression landscape captured in your scRNAseq experiments?

We appreciate this question. Our aim in this section was to validate broad cluster identities rather than reconstruct full temporal expression dynamics for each marker gene. We selected St.19 because it represents a stage at which all major head structures captured in the single-cell dataset are clearly established, making it an ideal reference point for cluster validation.

When temporal dynamics were central to the study's aims, specifically for CNCC and facial ectomesenchyme development, we performed HCR across multiple stages (Figs. 3-4). By contrast, extending cluster-validation HCR to all stages would primarily refine non-CNCC tissue dynamics and would not substantially affect the CNCC-focused conclusions.

Given the very limited availability of early catshark embryos, we prioritized multi-stage analyses for questions where developmental timing was essential and used a single representative stage where it was sufficient (solely to validate the annotation of non-CNCC clusters recovered in bioinformatic pipelines).

9. In the "CNCC-derived neuroglia are not detected in the scRNAseq analysis" section of the results, there is a grammatical error in the sentence that states "To investigate this cell lineage, we first examined the expression of PHOX2B, a canonical marker of autonomic neuronal (Fig. S8A)." This sentence should say autonomic neurons.

We thank the reviewer for noting this grammatical error and have corrected it.

10. In the same paragraph, there is a sentence stating, "In mice, late-migratory CNCC give rise to neuroglia of the cranial ganglia." It should be clarified that differentiation of the CNCC into neuroglial derivatives occurs sequentially after ectomesenchymal fate specification and if this is a conserved trait in non-mammalian species as well.

We agree and have revised this sentence to clarify that CNCC differentiation into neuroglial derivatives occurs sequentially after ectomesenchymal fate specification. We now explicitly note that this temporal hierarchy appears conserved in non-mammalian vertebrates (e.g., in zebrafish) (14).

11. "Nonetheless, the robust ALX1 expression in the catshark periocular mesenchyme supports its transcriptional equivalence to the frontonasal mesenchyme found in other Osteichthyes." This statement is a bit overreaching. Demonstrating functional equivalence between these two populations would provide the necessary evidence to support this claim.

We agree that the original wording was overreaching. We have toned down this statement and clarified that transcriptional similarity alone does not demonstrate functional equivalence, which will require future functional validation.

12. "EDNRB is expressed in early CNCC and becomes restricted to the PNS by St. 21." Was this statement

confirmed via coexpression of EDNRB with a marker of PNS cell types or is this conclusion purely observational?

This conclusion was based on careful examination of confocal stacks rather than formal co-expression analyses. EDNRB expression was consistently observed within cranial and dorsal root ganglia. If the opportunity to revise this work arises, we will map co-expression of EDNRB with markers of the PNS to confidently state that it is restricted to PNS structures at these stages. We will clarify this distinction in the revised text.

13. "The conserved expression patterns of the endothelin receptors in the catshark suggest functional conservation of this signaling pathway across vertebrates." This statement as written is overreaching. It should be clarified that this conclusion cannot be made without experiments confirming the function of each receptor.

We agree with this comment and have clarified the need for further functional validations to fully assess the presumed conserved role of the EDN pathway in Chondrichthyes.

14. In Figure 6, it is unclear how the classification of "mesenchymal condensations" versus "cartilage" is assigned to the images. Please clarify.

We apologize for the lack of clarity. We now explicitly reference the criteria used to distinguish mesenchymal condensations from cartilage in the Results section and point readers to the Methods, following previously established micro-CT-based criteria (17-19). Please see the lines 919-925.

References:

1. C. Fujimori, *et al.*, In vitro and in vivo gene transfer in the cloudy catshark *Scyliorhinus torazame*. *Development, Growth & Differentiation* 64, 558-565 (2022).
2. H. Jung, *et al.*, The Ancient Origins of Neural Substrates for Land Walking. *Cell* 172, 667-682.e15 (2018).
3. W. W. Ballard, J. Mellinger, H. Lechenault, A series of normal stages for development of *Scyliorhinus canicula*, the lesser spotted dogfish (Chondrichthyes: Scyliorhinidae). *Journal of Experimental Zoology* 267, 318-336 (1993).
4. Y. Lund-Ricard, A. Boutet, "Current Trends in Chondrichthyes Experimental Biology" in *Handbook of Marine Model Organisms in Experimental Biology*, (CRC Press, 2021).
5. J. A. Gillis, *et al.*, "Big insight from the little skate: *Leucoraja erinacea* as a developmental model system" in *Current Topics in Developmental Biology*, Emerging Model Systems in Developmental Biology., B. Goldstein, M. Srivastava, Eds. (Academic Press, 2022), pp. 595-630.
6. K. Onimaru, F. Motone, I. Kiyatake, K. Nishida, S. Kuraku, A staging table for the embryonic development of the brownbanded bamboo shark (*Chiloscyllium punctatum*). *Developmental Dynamics* 247, 712-723 (2018).
7. S. Menchero, *et al.*, Marsupial single-cell transcriptomics identifies temporal diversity in mammalian developmental programs. *Developmental Cell* 60, 3339-3356.e5 (2025).
8. R. Soldatov, *et al.*, Spatiotemporal structure of cell fate decisions in murine neural crest. *Science* 364, eaas9536 (2019).
9. L. Adams, *et al.*, Monitoring egg fertility, embryonic morbidity, and mortality in an oviparous elasmobranch using ultrasonography. *Front. Vet. Sci.* 11 (2024).
10. F. Santagati, F. M. Rijli, Cranial neural crest and the building of the vertebrate head. *Nat Rev Neurosci* 4, 806-818 (2003).

11. M. L. Martik, M. E. Bronner, Riding the crest to get a head: neural crest evolution in vertebrates. *Nat Rev Neurosci* **22**, 616-626 (2021).
12. V. A. Sleight, J. A. Gillis, Embryonic origin and serial homology of gill arches and paired fins in the skate, *Leucoraja erinacea*. *eLife* **9**, e60635 (2020).
13. S. Oulion, *et al.*, Evolution of repeated structures along the body axis of jawed vertebrates, insights from the *Scyliorhinus canicula* Hox code. *Evol Dev* **13**, 247-259 (2011).
14. D. Tatarakis, *et al.*, Single-cell transcriptomic analysis of zebrafish cranial neural crest reveals spatiotemporal regulation of lineage decisions during development. *Cell Rep* **37**, 110140 (2021).
15. Z. Johanson, C. Boisvert, A. Maksimenko, P. Currie, K. Trinajstić, Development of the Synarcual in the Elephant Sharks (Holocephali; Chondrichthyes): Implications for Vertebral Formation and Fusion. *PLOS ONE* **10**, e0135138 (2015).
16. S. Kuratani, N. Horigome, Developmental Morphology of Branchiomic Nerves in a Cat Shark, *Scyliorhinus torazame*, with Special Reference to Rhombomeres, Cephalic Mesoderm, and Distribution Patterns of Cephalic Crest Cells. *Jzoo* **17**, 893-909 (2000).
17. M. Kaucka, *et al.*, Oriented clonal cell dynamics enables accurate growth and shaping of vertebrate cartilage. *eLife* **6**, e25902 (2017).
18. M. Kaucka, *et al.*, Signals from the brain and olfactory epithelium control shaping of the mammalian nasal capsule cartilage. *eLife* **7**, e34465 (2018).
19. M. Tesařová, *et al.*, Use of micro computed-tomography and 3D printing for reverse engineering of mouse embryonal nasal capsule. *J. Inst.* **11**, C03006 (2016).

Rebuttal decision

Dear Dr Kaucka Petersen,

Thank you for your recent rebuttal letter. I understand how disappointed you must feel.

Given the opinions stated by the reviewers, I had no choice but to reject the paper.

However, we are always willing to give authors the chance to defend their manuscripts. In the light of the comments you make in your letter, I have decided to proceed as follows. I would be happy for you to submit a revised version of your manuscript, but it would need to be submitted as a Techniques and Resources Article (please see information on this article type here: <https://journals.biologists.com/dev/pages/article-types#techniques>).

Our Admin Team has changed your article type to Techniques and Resources Article so it is ready for you to submit your revisions, however, if you wish to not proceed with this new article type and instead submit elsewhere, please contact the Admin Team at dev@biologists.com as soon as possible.

To submit a revised manuscript, please go to: <https://www.editorialmanager.com/develop/> and click on the 'Submissions Needing Revision' within the Author Main Menu.

With best wishes,
Dr. Cassandra G. Extavour

First revision

Author response to reviewers' comments

Response to the Reviewers

We thank the Reviewers for their careful and thoughtful evaluation of our manuscript and for their constructive suggestions. We have revised the manuscript accordingly and respond to each comment below (in blue).

Reviewer #1

In this beautifully presented manuscript, the authors present single-cell transcriptomic data from the head region (otic level and above) of the catshark. They then focus on the cranial neural crest subset of cells. They validate expression of several genes by immunofluorescence and/or hybridization chain reaction in situ over developmental time. In addition, they use synchrotron tomography to examine anatomical changes. Their findings indicate that the cranial neural crest of the catshark is largely conserved compared to other gnathostomes but identify subtle lineage-specific adaptations. Overall, while this represents a good start and the paper is nicely done given the data in hand, it lacks novel insights and functional validation of novel candidate genes is missing. The following suggestions would improve the manuscript and make it a good candidate for publication in Development.

We would like to cordially thank Reviewer #1 for their positive assessment of the manuscript and for their valuable comments and suggestions, which helped us improve the clarity, limitations, and interpretation of the findings.

Major comments:

6. The authors dissected and dissociated the whole head region above the otic level (Fig. 2A), which encompasses a wide range of cell types of diverse embryonic origin, including placode-, mesoderm-, endoderm- and non-neural ectoderm-derived cells. As such, the "neural crest" cluster represents a very small fraction of the dataset. Although technically challenging, given that electroporation is feasible in the catshark (Fujimori et al., 2022), the authors should attempt to repeat their single-cell transcriptomic analysis using pure populations of cranial neural crest cells. These could be obtained by injection followed by electroporation of conserved enhancers (e.g., Sox10 or FoxD3 enhancers; e.g. Hockman et al., 2019 or Parker et al., 2019) that function cross species. This approach would provide a much deeper and more interpretable dataset.

We appreciate the reviewer's thoughtful suggestion to enrich cranial neural crest cells (CNCC) using electroporation-based genetic labeling. We agree that such approach would be extremely valuable if technically feasible. However, we would like to clarify that this methodology is not currently applicable to early-stage chondrichthyan embryos, particularly at the stages when CNCC specification, delamination, and early migration occur.

To date, electroporation in chondrichthyans has only been reported in adult animals or late-stage embryos (e.g., St. 31-32 in the cloudy catshark; Fujimori et al., 2022) (1), and more recently, in St. 27-28 little skate embryos (2). To our knowledge, these reports represent the entirety of the published literature on electroporation in this phylogenetic lineage *in vivo*, and no study has achieved electroporation at early embryonic stages comparable to those analyzed in our study (St. 16-20).

Early chondrichthyan embryos differ fundamentally from later stages: they cannot be cultured outside the egg case, are extremely sensitive to environmental perturbations, and do not survive the manipulations associated with electroporation (3-6). Even minimal disturbance of the yolk

sac is typically lethal at these early stages. By contrast, embryos from St.26 onward are substantially more resilient, consistent with the stages used in the published studies cited above (3, 5, 6). In a recent book chapter on non-model organisms by leaders in the field, it is explicitly noted that experimental manipulation of early chondrichthyan embryos remains under development and lacks robust, reproducible protocols (5).

In addition, the enhancers suggested by the reviewer (e.g., Sox10, FoxD3) have never been functionally validated *in vivo* in chondrichthyans, precisely because early-stage genetic manipulation is not yet achievable. Establishing enhancer-driven labeling at these stages would therefore require extensive method development and validation, constituting an independent research program well beyond the scope of the present study.

From a biological perspective, we also note that enriching exclusively for CNCC would eliminate important contextual information. Our whole-head strategy enabled us to analyze interactions between CNCC and their surrounding tissues, including mesoderm, CNS, and ectoderm, as demonstrated by the presented CellChat analysis (Fig. 3F). These cross-tissue signaling relationships would not be accessible in a CNCC-only dataset. Importantly, the proportion of CNCC recovered reflects their *in vivo* abundance at these stages.

This approach is consistent with recent single-cell studies in other systems. For example, whole-embryo profiling of marsupial CNCC recovered early, migratory and ectomesenchymal populations without sensory or autonomic derivatives at comparable stages, in agreement with our findings (7). Even genetically enriched CNCC datasets, such as Soldatov et al. (Science, 2019) (8), recovered similar numbers of CNCC at equivalent developmental stages (approximately 500 at mouse E8.5 versus 353 in our dataset). We further complement our single-cell data with extensive spatial validation (HCR, immunofluorescence) and synchrotron-based 3D reconstructions spanning later developmental stages, providing a more comprehensive picture of developmental dynamics.

Finally, we emphasize the severe practical constraints of working with chondrichthyan embryos. Catsharks produce very limited numbers of embryos, with strong seasonal variation and high embryonic mortality and morbidity (3, 5, 6, 9). These constraints make repetition or expansion of early-stage single-cell experiments exceptionally challenging and explain why such datasets remain exceedingly rare in this lineage. To the best of our knowledge, this is the first early developmental dataset from any chondrichthyan species.

Taken together, while we fully agree with the conceptual value of the reviewer's suggestion and we would be generally extremely enthusiastic about the opportunity to perform the first-ever (successful) electroporation and genetic tracing in early embryos of the small spotted catshark, the proposed experiment is currently not feasible in chondrichthyans. We respectfully maintain that our whole-head single-cell strategy provides a robust, biologically meaningful, and unique dataset that is well-aligned with the current technical state of the field.

7. The authors have not included the posterior cranial neural crest, which primarily populates the branchial arch region. This omission is problematic, as this head region represents an important component of the cranial neural crest. Therefore, the rationale for excluding it is unclear—particularly given that the title and scope of the study refer to the "cranial neural crest", rather than only the pre-otic subpopulations (mandibular and hyoid). Furthermore, in the dissections of later developmental stages (19 and 20), the hyoid arch derived from the pre-otic cranial neural crest appears to be missing. Is this indeed the case, or does the dashed line in Fig. 2A encompass this region?

We thank the reviewer for highlighting the diversity of CNCC populations. As correctly noted, CNCC are subdivided into trigeminal (mandibular), hyoid, and branchial streams, each contributing to distinct cranial structures. Trigeminal CNCC generate the first pharyngeal arch, facial skeleton, and anterior cranial ganglia; hyoid CNCC contribute to second arch derivatives and components of the middle ear; and branchial CNCC populate arches 3-7 in the catshark, forming structures such as the gill cartilage and posterior cranial ganglia (10-12). While branchial CNCC indeed represent an essential component of the head, they do not contribute to the development of facial structures.

Our study specifically focuses on facial evolution, which is generated exclusively by the trigeminal CNCC stream. For this reason, our single-cell analysis and synchrotron-based reconstructions were deliberately restricted to the pre-otic region. This is a conceptual decision aligned with the biological question under investigation, rather than an omission.

We respectfully suggest that changing the title from “cranial neural crest” to “pre-otic neural crest” would introduce unnecessary confusion for the readers; the term “cranial neural crest” is conventionally used to describe all anterior NCC streams (8, 10, 11), and our revised manuscript explicitly explains why the trigeminal CNCC are the focus of the study and the core of the evolutionary question.

Importantly, the study does not exclude hyoid or branchial CNCC from morphological or molecular analyses. All imaging-based datasets (immunofluorescence and HCR) encompass the entire embryonic head, including all CNCC streams, and our descriptions of delamination, early migration (Fig. 1), and gene expression (Fig. 5) include the trigeminal, hyoid, and branchial populations.

Regarding the reviewer’s concern about potential inclusion of hyoid CNCC in the single-cell dataset at St.19-20, we screened for *HOX* gene expression, which reliably marks hyoid and branchial CNCC in the catshark (13). This analysis, now included as a new Supplementary Figure (Fig. S2), show that *HOX* expression is confined to hindbrain-associated clusters and absent from the CNCC cluster, confirming that the dataset contains only trigeminal CNCC. The dashed boundaries in Fig. 2A therefore accurately reflect the dissected region, and the hyoid population is not represented in the single-cell dataset.

In summary, the exclusion of posterior CNCC from the single-cell dataset was a deliberate decision to align with the scope of the study. The morphological and gene expression analyses encompass all CNCC streams, ensuring that the developmental context of the trigeminal population is adequately represented. We have clarified this rationale more explicitly in the revised manuscript to avoid confusion.

8. Neural crest cells give rise to all of the glia of the peripheral nervous system. Why, then, is this neural crest derivative absent from the dataset? Although the authors dedicate a subsection to this topic, it warrants further validation given its biological importance, including the potential delayed emergence of this neural crest derivative. In its current form, the apparent absence of neuroglia further suggests that the cranial neural crest population is underrepresented in the current analysis.

We agree that CNCC-derived neuroglia represent an important lineage. Their absence from the single-cell dataset reflects developmental timing, not underrepresentation of CNCC.

The presented scRNA-seq dataset spans St.16-20, a window during which CNCC undergo delamination, migration, and early differentiation toward ectomesenchymal fates. In vertebrates, neuroglial derivatives arise later than ectomesenchymal lineages, a temporal hierarchy that has been documented in multiple species (7, 8, 14).

Consistent with this, we performed *PHOX2B* HCR analysis and found that the first CNCC-derived neuroglia appear at St.21-22, outside the window of our single-cell sampling. Thus, their absence from the dataset is expected and biologically informative rather than indicative of technical bias. We note that similar stage- dependent absences are common in developmental single-cell studies, including recent studies, such as marsupial CNCC atlas (7). Similarly to our work, the marsupial NCC dataset did not recover sensory-lineage derivatives, not because opossum CNCC fail to produce them, but because the latest stage analyzed (E10.5) precedes their emergence (7). Our dataset reflects the same principle.

The embryonic stages we selected for single-cell profiling were guided by SOX9 immunofluorescence (Fig. 1) and morphological comparisons with other vertebrate species; however, the emergence of neuroglia at St.21-22 could not have been predicted beforehand. Such discoveries are a natural (and valuable) outcome of exploratory developmental studies and point to exciting avenues for future research, as acknowledged in our Discussion. Importantly, the timing of neuroglial emergence in chondrichthyans has not been previously documented. Our study, therefore, provides the first

empirical evidence that places this transition at St. 21-22.

We have clarified this point and expanded the discussion of developmental timing and the chapter “limitations” in the revised manuscript.

9. While the authors focus on shared neural crest traits across gnathostomes, they also point to morphological differences. Are there candidate genes that reflect these differences? Highlighting and discussing such genes would strengthen the study and provide valuable insight into lineage-specific adaptations.

Thank you for this interesting question and suggestion. In response, we have further explored the periostin (POSTN) signaling pathway, which was identified by the CellChat analysis (Fig. 3F) as a strong signal secreted by the notochord and received by the catshark CNCC/ectomesenchyme. To our knowledge, this signaling pathway has not been previously reported to play a role in CNCC biology in other species. Moreover, it was not identified in a similar murine single-cell CellChat analysis on embryonic facial morphogenesis (15), suggesting species-specific differences.

First, we searched for available single-cell transcriptomics datasets of developing gnathostome species to gain insight into the cellular sources of *POSTN* during embryogenesis. We screened for *POSTN*-expressing cell types and tissues in developing mouse (*Mus musculus*), chicken (*Gallus gallus*), Western clawed frog (*Xenopus tropicalis*) and zebrafish (*Danio rerio*) (Table S1) (16-21). In these single-cell transcriptomic datasets, *POSTN* is expressed in the notochord in chicken and Western clawed frog, while in mouse and zebrafish *POSTN* is mainly expressed in mesodermal tissues, ectodermal cells, and the gut (Table S1).

To further validate the single-cell data, we performed HCR for *POSTN* in catshark, mouse, and chicken embryos and compared their expression patterns (Fig. 4). The HCR results corroborated the conclusions drawn from the single-cell data: *POSTN* is strongly expressed in the notochord in catshark and chicken but is absent from this embryonic structure in mouse. These findings suggest a potential differential use of the *POSTN* signalling pathway in mammals and Actinopterygii compared to other gnathostome lineages and highlight the importance of cross-tissue communication in CNCC biology and evolution.

We have expanded the Results to emphasize the *POSTN* signaling pathway as a candidate contributor to species-specific craniofacial patterning, while clearly framing this as a hypothesis that will require future functional investigation.

10. The study currently lacks functional testing of any genes identified by single-cell RNA-seq. Although I do understand that working with non-traditional model organisms represents significant technical challenges, and CRISPR-mediated loss-of-function is not yet established in sharks or skates, functional assays remain feasible. For instance, in vivo morpholino-mediated knockdowns could be employed to assess gene function, with resulting phenotypes analyzed using synchrotron tomography. Testing a few candidate genes in this manner would provide valuable insights whether gene functions are conserved or divergent across species, and would substantially strengthen the study, bringing it closer to the standards expected for publication in a journal such as Development.

We fully agree that functional assays would greatly advance understanding of gene function in chondrichthyan CNCC development. However, gene-specific functional perturbation has not yet been achieved in chondrichthyan embryos, including morpholino-mediated knockdown.

To our knowledge, no published study has demonstrated successful morpholino delivery or efficacy in sharks or skates, and the same biological and technical constraints that limit CRISPR approaches also apply to morpholinos. Early embryos are not amenable to one-cell injection, genetic manipulation, or prolonged *ex vivo* culture. As such, the suggested experiments are not currently feasible.

At present, functional approaches in early chondrichthyan embryos are mostly limited to fate mapping using lipophilic dyes and broad pharmacological perturbations (5), which do not permit gene-specific or tissue-specific loss-of-function analyses. We have clarified these constraints in the

Discussion.

We certainly do share the reviewer's enthusiasm for future methodological advances in this area, but emphasize that the absence of functional assays reflects field-wide technical limitations, not an omission specific to this study. We cordially thank the Reviewer for their understanding.

11. The title does not reflect the content of the manuscript as the paper profiles part of the head and the neural crest component is a minor one.

We respectfully disagree. The study is explicitly centered on CNCC and their role in facial evolution, as stated in the title. While CNCC constitute a subset of the single-cell dataset, they do form the conceptual and biological focus of the manuscript. The study integrates scRNA-seq with extensive spatial and morphological analyses, focusing on CNCC specification, migration, signaling interactions, and their contribution to facial morphogenesis.

Importantly, the manuscript presents a comprehensive analysis of CNCC beyond their representation in the single-cell dataset. We provide detailed descriptions of CNCC specification, delamination, and migratory trajectories, as well as inference of signaling interactions between CNCC and surrounding cranial tissues. In addition, the study includes extensive gene expression mapping that encompass all CNCC subpopulations, not solely the trigeminal stream highlighted in the single-cell experiment, and synchrotron-based 3D reconstructions that illuminate facial morphogenesis.

Thus, although the single-cell component necessarily concentrates on trigeminal CNCC, the broader manuscript examines CNCC biology at multiple levels and across multiple methodologies. For these reasons, we believe that the current title accurately reflects the scope and emphasis of the work.

Minor comments:

All minor comments from Reviewer #1 have been addressed as suggested. Figures have been updated, text has been clarified or toned down where appropriate, and overstatements have been corrected. Specific changes are detailed in the revised manuscript.

12. I recommend removing the reference to the "accelerated hyoid stream" in Fig. 7, as zebrafish do not possess an accelerated hyoid neural crest. If the authors wish to base their schematic on zebrafish cranial neural crest migration, this statement should either be removed or appropriately clarified in the text. The accelerated hyoid neural crest may instead represent a lineage-specific adaptation present only in certain fish lineages.

In response to this comment, we have modified the figure and clarified the information in the main text and the corresponding figure legend. Please see the adapted Figure 7 (now Fig. 8) and its respective legend.

13. Although cartilaginous fishes occupy a key phylogenetic position as the closest living branch to the divergence between jawed and jawless vertebrates, the extant Chondrichthyes are themselves highly derived from their fossil ancestors. Therefore, I suggest tempering the statement in the Abstract: "little is known about CNCC properties in cartilaginous fishes (Chondrichthyes), leaving the ancestral condition of the first jawed vertebrates unresolved." The ancestral condition could instead be more appropriately inferred through comparative analyses among jawless vertebrates, Chondrichthyes, and the deepest extant branches of Osteichthyes.

We agree with your comment and have thus removed this statement from the abstract.

Closing remark:

We believe that the revised manuscript addresses the reviewers' concerns while clearly articulating the current technical boundaries of developmental research in chondrichthyans. Importantly,

the study leverages the most advanced and reliable methodologies currently available for early-stage shark embryos, combining single-cell transcriptomics with high-resolution spatial validation and synchrotron-based 3D reconstruction to extract maximal biological insight from a system where experimental manipulation remains extremely limited.

Historically, the journal *Development* has played a central role in advancing rigorous, conceptually driven developmental biology in non-traditional model organisms by carefully valuing executed descriptive and comparative studies, provided technical constraints are transparently acknowledged, and conclusions are appropriately framed. We believe our work aligns well with this tradition, as it provides a foundational molecular and morphological framework for cranial neural crest development in chondrichthyans, motivates future methodological and functional studies, and establishes reference data that will be essential once new experimental tools become available.

For these reasons, we respectfully resubmit with the conviction that the manuscript meets the rigorous standards of transparency and conceptual contribution expected for publication in *Development*.

Upon discussion with the handling editor, we have been invited to resubmit the revised manuscript as a Techniques and Resources Article. We believe that the work and datasets presented here will constitute a valuable resource for the study of shark embryonic development, as well as for investigations into the development and evolution of neural crest cells. Furthermore, this study establishes and promotes the broader application of the presented techniques in chondrichthyans. In addition, we have developed an information-rich, open-access, interactive database integrating the generated single-cell transcriptomic data and 3D imaging datasets, enabling rapid, efficient, and user-friendly access to and dissemination of these resources: <https://www.evolbio.mpg.de/escamilla-sharknc> (access to the database will be enabled upon publication).

Reviewer #2

This manuscript focuses on characterizing development of cranial neural crest cells in the chondrichthyan, *Scyliorhinus canicula* (small-spotted catshark), to evaluate the extent of conservation of these mechanisms in the sister group to Osteichthyes and uncover insights into the development of diverse craniofacial morphology among gnathostomes (jawed vertebrates). The question being posed is interesting, as several papers have focused related questions on other jawed species including bony fish, but much less is known about neural crest cells in cartilaginous fishes. The researchers used molecular, genetic, and morphological approaches to substantiate the conservation of core developmental gene programs for early CNCC development in their model organism. They also show that diverse craniofacial morphology in Chondrichthyes is likely underscored by divergent behavior of ectomesenchymal progenitors during early embryogenesis. This research will provide valuable information to the fields of neural crest cell biology and EvoDevo as it illuminates potential novel mechanisms driving craniofacial development. It is also a valuable resource for studying early embryogenesis at the molecular level in a non-traditional model organism.

We thank Reviewer #2 for the positive and encouraging assessment of our study and for the constructive suggestions.

All minor comments from Reviewer #2 have been addressed. Figures have been annotated for clarity, text has been revised to improve precision and consistency, and limitations regarding species representation and functional inference have been explicitly acknowledged.

Minor comments:

7. It is not clear why the top row in figure 1 is included because the higher mag or close up images in the figure (G) are such better resolution. Row A seems like it could be placed in supplemental without much loss to the manuscript.

We thank the reviewer for this suggestion. We included the top row of Figure 1 (panel A) to provide a developmental and anatomical overview across multiple intermediate stages that cannot be easily conveyed by high-magnification images alone. These lower-magnification views allow readers to appreciate the spatial relationship between cranial and trunk NCC populations and to contextualize the higher-resolution panels.

In addition, Fig. 1A serves an important organizational purpose: it contains the dashed reference lines indicating the positions of panels B-F, while Fig. 1G labels key anatomical structures. Including all information within a single high-magnification panel would substantially reduce clarity. For these reasons, we believe that retaining panel A improves readability and accessibility, particularly for readers less familiar with catshark embryonic anatomy.

8. Line 17 says that primary neurulation occurs in "primary neurulation, a process observed in amniotes, bichirs, and sterlets," but it is not clear why they limit this to these species? Several anamniotes including amphibians (frogs and axolotls) and zebrafish (Werner et al., 2021) use primary neurulation as well.

We thank the reviewer for pointing out this imprecision. We agree that the original wording was unnecessarily restrictive and inadvertently excluded several anamniote species in which primary neurulation also occurs, including amphibians and zebrafish. We have revised the sentence accordingly.

9. In the introduction, there is mention of "shark-specific facial morphology" and in the section of the results titled "The periocular ectomesenchyme of the catshark is homologous to the frontonasal ectomesenchyme in Osteichthyes" a similar phrase is used: "characteristic Chondrichthyes facial morphology." These statements are in direct conflict with a claim made in the discussion section stating "Chondrichthyes display an impressive variety of facial morphologies..." The authors should clarify why the small-spotted catshark is a representative species of the chondrichthyans that is beneficial to study, while also addressing the limitations associated with drawing conclusions about an entire phylogenetic group based on a single species.

We appreciate this thoughtful observation and the opportunity to clarify our intent. Our references to "shark-specific" or "characteristic chondrichthyan" facial morphology were meant to highlight adult morphological outcomes that ultimately derive from CNCC, thereby emphasizing the evolutionary relevance of studying facial ectomesenchyme at the embryonic level.

By contrast, our discussion of the impressive diversity of chondrichthyan facial morphologies refers explicitly to adult forms across the clade, which indeed show substantial variation. These statements are therefore not contradictory but refer to morphology at different developmental stages. As in other vertebrate groups, embryos within a lineage are far more similar to one another during early development than their adult anatomies might suggest.

At the embryonic stages analyzed here, available data from multiple chondrichthyan species (including small-spotted catshark, cloudy catshark, little skate, and elephant shark) indicate a high degree of conservation in the organization of periocular and facial ectomesenchyme (12, 22, 23). This supports the use of the small-spotted catshark as a representative and experimentally accessible model for early chondrichthyan development, while fully acknowledging that no single species can capture the full morphological diversity of an entire clade.

We have expanded the Discussion to clarify this distinction and to explicitly acknowledge the limitations associated with drawing broader evolutionary inferences from a single species.

10. An overall observation for the main figures: arrowheads, asterisks, or some other annotation should be added in the main figures to highlight key characteristics of the images discussed by the authors as it is sometimes unclear what feature in the images is being discussed in the text. This could significantly improve clarity of the results.

We fully agree and thank the reviewer for this helpful suggestion. We have added annotations throughout the main and supplementary figures to more clearly highlight the features discussed in the text and improve figure readability.

11. There seems to be a discrepancy between the data for MDGA2 in the dot plot in Figure 3C and the expression pattern of MDGA2 in Fig. 3D- specifically- in the dot plot it appears to be groups with early CNCC but in the expression image- it appears to be late and distinct from the UNC5C expression which mirrors Sox9 from figure 1. This is an example of where arrowheads or asterisks could be useful to clarify the relevant regions of expression (or absence of expression) that are discussed in the text. Sections should also be included if they show the co-localization better.

We thank the reviewer for drawing attention to this point and agree that the original presentation could be confusing. The “early CNCC” cluster encompasses a continuum of CNCC states, including non- delaminating, delaminating, and migrating cells, which are not resolved into distinct subclusters at the chosen resolution.

As a result, although both *UNC5C* and *MDGA2* mark this cluster, they are expressed in different transitional CNCC subpopulations. To avoid misinterpretation, we have removed this panel and revised the text accordingly.

12. Figure 5 could benefit from more annotations as well, but the images are beautiful.

We thank the reviewer for the positive feedback and have added additional annotations to Figure 5 to improve clarity.

13. There are several sentences at the end of the second paragraph of the results section titled “CNCC migratory streams are conserved in the catshark” that are repetitive with earlier information in that same paragraph. These sentences, which focus on the hyoid and branchial CNCC migratory streams, could be removed or condensed to improve clarity.

We agree with the reviewer and have condensed this paragraph by removing repetitive sentences, improving clarity and flow.

14. “To validate cluster identities, we performed whole-mount HCR in situ hybridization using St. 19 embryos as a representative stage for the dataset.” What is the justification for using a single stage here rather than several stages? Is this sufficient to validate the complex expression landscape captured in your scRNAseq experiments?

We appreciate this question. Our aim in this section was to validate broad cluster identities rather than reconstruct full temporal expression dynamics for each marker gene. We selected St.19 because it represents a stage at which all major head structures captured in the single-cell dataset are clearly established, making it an ideal reference point for cluster validation.

When temporal dynamics were central to the study’s aims, specifically for CNCC and facial ectomesenchyme development, we performed HCR across multiple stages (Figs. 3, 5). By contrast, extending cluster-validation HCR to all stages would primarily refine non-CNCC tissue dynamics and would not substantially affect the CNCC-focused conclusions.

Given the very limited availability of early catshark embryos, we prioritized multi-stage analyses for questions where developmental timing was essential and used a single representative stage where it was sufficient (solely to validate the annotation of non-CNCC clusters recovered in bioinformatic pipelines).

15. In the “CNCC-derived neuroglia are not detected in the scRNAseq analysis” section of the results, there is a grammatical error in the sentence that states “To investigate this cell lineage, we first examined the expression of PHOX2B, a canonical marker of autonomic neuronal (Fig. S8A).” This sentence should say autonomic neurons.

We thank the reviewer for noting this grammatical error and have corrected it.

16. In the same paragraph, there is a sentence stating, “In mice, late-migratory CNCC give rise to neuroglia of the cranial ganglia.” It should be clarified that differentiation of the CNCC into neuroglial derivatives occurs sequentially after ectomesenchymal fate specification and if this is a

conserved trait in non- mammalian species as well.

We agree and have revised this sentence to clarify that CNCC differentiation into neuroglial derivatives occurs sequentially after ectomesenchymal fate specification. We now explicitly note that this temporal hierarchy appears conserved in non-mammalian vertebrates (e.g., in zebrafish) (14).

17. "Nonetheless, the robust ALX1 expression in the catshark periocular mesenchyme supports its transcriptional equivalence to the frontonasal mesenchyme found in other Osteichthyes." This statement is a bit overreaching. Demonstrating functional equivalence between these two populations would provide the necessary evidence to support this claim.

We agree that the original wording was overreaching. We have toned down this statement and clarified that transcriptional similarity alone does not demonstrate functional equivalence, which will require future functional validation.

18. "EDNRB is expressed in early CNCC and becomes restricted to the PNS by St. 21." Was this statement confirmed via coexpression of EDNRB with a marker of PNS cell types or is this conclusion purely observational?

This conclusion was initially based on careful examination of confocal stacks in which *EDNRB* expression was consistently observed within the PNS (cranial and dorsal root ganglia). To verify this, we mapped co- expression of *EDNRB* with *NEUROD1*, a known marker of peripheral ganglia (24). Expression of both *EDNRB* and *NEUROD1* was observed within the PNS, although they seem to label distinct cellular subpopulations. Nonetheless, this confirms that *EDNRB* is expressed within the PNS at this developmental stage in the small-spotted catshark. This information is now included in Fig. S10B.

19. "The conserved expression patterns of the endothelin receptors in the catshark suggest functional conservation of this signaling pathway across vertebrates." This statement as written is overreaching. It should be clarified that this conclusion cannot be made without experiments confirming the function of each receptor.

We agree with this comment and have clarified the need for further functional validations to fully assess the presumed conserved role of the EDN pathway in Chondrichthyes.

20. In Figure 6, it is unclear how the classification of "mesenchymal condensations" versus "cartilage" is assigned to the images. Please clarify.

We apologize for the lack of clarity. We now explicitly reference the criteria used to distinguish mesenchymal condensations from cartilage in the Results section and point readers to the Methods, following previously established micro-CT-based criteria (25-27). Please see the lines 882-888.

References:

3. C. Fujimori, *et al.*, In vitro and in vivo gene transfer in the cloudy catshark *Scyliorhinus torazame*. *Development, Growth & Differentiation* 64, 558-565 (2022).
4. H. Jung, *et al.*, The Ancient Origins of Neural Substrates for Land Walking. *Cell* 172, 667-682.e15 (2018).
5. W. W. Ballard, J. Mellinger, H. Lechenault, A series of normal stages for development of *Scyliorhinus canicula*, the lesser spotted dogfish (Chondrichthyes: Scyliorhinidae). *Journal of Experimental Zoology* 267, 318-336 (1993).
6. Y. Lund-Ricard, A. Boutet, "Current Trends in Chondrichthyes Experimental Biology" in *Handbook of Marine Model Organisms in Experimental Biology*, (CRC Press, 2021).
7. J. A. Gillis, *et al.*, "Big insight from the little skate: *Leucoraja erinacea* as a developmental model

- system” in *Current Topics in Developmental Biology*, Emerging Model Systems in Developmental Biology., B. Goldstein, M. Srivastava, Eds. (Academic Press, 2022), pp. 595-630.
8. K. Onimaru, F. Motone, I. Kiyatake, K. Nishida, S. Kuraku, A staging table for the embryonic development of the brownbanded bamboo shark (*Chiloscyllium punctatum*). *Developmental Dynamics* **247**, 712-723 (2018).
 9. S. Menchero, *et al.*, Marsupial single-cell transcriptomics identifies temporal diversity in mammalian developmental programs. *Developmental Cell* **60**, 3339-3356.e5 (2025).
 10. R. Soldatov, *et al.*, Spatiotemporal structure of cell fate decisions in murine neural crest. *Science* **364**, eaas9536 (2019).
 11. L. Adams, *et al.*, Monitoring egg fertility, embryonic morbidity, and mortality in an oviparous elasmobranch using ultrasonography. *Front. Vet. Sci.* **11** (2024).
 12. F. Santagati, F. M. Rijli, Cranial neural crest and the building of the vertebrate head. *Nat Rev Neurosci* **4**, 806-818 (2003).
 13. M. L. Martik, M. E. Bronner, Riding the crest to get a head: neural crest evolution in vertebrates. *Nat Rev Neurosci* **22**, 616-626 (2021).
 14. V. A. Sleight, J. A. Gillis, Embryonic origin and serial homology of gill arches and paired fins in the skate, *Leucoraja erinacea*. *eLife* **9**, e60635 (2020).
 15. S. Oulion, *et al.*, Evolution of repeated structures along the body axis of jawed vertebrates, insights from the *Scyliorhinus canicula* Hox code. *Evol Dev* **13**, 247-259 (2011).
 16. D. Tatarakis, *et al.*, Single-cell transcriptomic analysis of zebrafish cranial neural crest reveals spatiotemporal regulation of lineage decisions during development. *Cell Rep* **37**, 110140 (2021).
 17. A. P. Murillo-Rincón, *et al.*, Positional programs in early murine facial development and their role in human facial shape variability. *Nat Commun* **16**, 10112 (2025).
 18. X. Ibarra-Soria, *et al.*, Defining murine organogenesis at single-cell resolution reveals a role for the leukotriene pathway in regulating blood progenitor formation. *Nat Cell Biol* **20**, 127-134 (2018).
 19. B. Pijuan-Sala, *et al.*, A single-cell molecular map of mouse gastrulation and early organogenesis. *Nature* **566**, 490-495 (2019).
 20. T. Rito, *et al.*, Timely TGF β signalling inhibition induces notochord. *Nature* **637**, 673-682 (2025).
 21. J. A. Briggs, *et al.*, The dynamics of gene expression in vertebrate embryogenesis at single-cell resolution. *Science* **360**, eaar5780 (2018).
 22. M. Lange, *et al.*, A multimodal zebrafish developmental atlas reveals the state-transition dynamics of late-vertebrate pluripotent axial progenitors. *Cell* **187**, 6742-6759.e17 (2024).
 23. D. E. Wagner, *et al.*, Single-cell mapping of gene expression landscapes and lineage in the zebrafish embryo. *Science* **360**, 981-987 (2018).
 24. Z. Johanson, C. Boisvert, A. Maksimenko, P. Currie, K. Trinajstić, Development of the Synarcual in the Elephant Sharks (Holocephali; Chondrichthyes): Implications for Vertebral Formation and Fusion. *PLOS ONE* **10**, e0135138 (2015).
 25. S. Kuratani, N. Horigome, Developmental Morphology of Branchiomic Nerves in a Cat Shark, *Scyliorhinus torazame*, with Special Reference to Rhombomeres, Cephalic Mesoderm, and Distribution Patterns of Cephalic Crest Cells. *Jzoo* **17**, 893-909 (2000).

26. P. O'Neill, R. B. McCole, C. V. H. Baker, A molecular analysis of neurogenic placode and cranial sensory ganglion development in the shark, *Scyliorhinus canicula*. *Dev Biol* **304**, 156-181 (2007).
27. M. Kaucka, *et al.*, Oriented clonal cell dynamics enables accurate growth and shaping of vertebrate cartilage. *eLife* **6**, e25902 (2017).
28. M. Kaucka, *et al.*, Signals from the brain and olfactory epithelium control shaping of the mammalian nasal capsule cartilage. *eLife* **7**, e34465 (2018).
29. M. Tesařová, *et al.*, Use of micro computed-tomography and 3D printing for reverse engineering of mouse embryonal nasal capsule. *J. Inst.* **11**, C03006 (2016).

Second decision letter

MS ID#: dev.205258R1

MS TITLE: Developmental dynamics of catshark cranial neural crest cells provide insights into gnathostome facial evolution

AUTHORS: Marketa Kaucka Petersen; Elio Escamilla-Vega; Andrea P. Murillo-Rincón; Louk W. G. Seton; Ann-Katrin Koch; Stella Kyomen; Carsten Fortmann-Grote; Jörg U. Hammel; Timo Moritz
Article Type: Techniques and Resources Article

Dear Dr Kaucka Petersen,

I am happy to tell you that your manuscript has been accepted for publication in *Development*, pending our standard publication integrity checks.

Reviewer 1

The revised version of the manuscript has been significantly improved, and I greatly appreciate the authors' comprehensive responses to all of my comments. While I initially had concerns that led me to recommend substantial revisions, I would like to clarify that all of my comments and suggestions were intended to increase the impact of this otherwise very nice study. Since the authors have now addressed the major issues appropriately and clearly explained the technical limitations of what could be done, and given that the manuscript will be published as a Techniques and Resources Article, I believe that the manuscript is now suitable for publication.

I would also like to highlight the authors' detailed rebuttal, which clearly explained the difficulties associated with shark embryology.

I agree that CellChat analysis can help suggest potential tissue or cell-cell interactions; however, the analysis alone cannot serve as functional validation of those interactions, as authors know. Nevertheless, given the nature of this study and the absence of feasible functional experiments, the authors have used this model appropriately and interpreted the results with reasonable caution.

I appreciate the authors' decision to elaborate further on some potential candidates, such as postn expression in the notochord, which improves the manuscript. The notochord is a very important signaling center in the developing embryo, and therefore it is not surprising that it may represent one of the potential signaling sources for developing head. Is the absence of postn in the murine dataset related to the much larger craniofacial region (the "new head") compared with so-called lower vertebrates? This is a very interesting observation, and some level of speculation or discussion could be added to the manuscript.

Overall, the manuscript now presents a useful resource with improved clarity and appropriate interpretation of its analyses. I support publication in its current revised form.

Reviewer 2

SUMMARY OF THE ADVANCE MADE IN THIS PAPER AND ITS POTENTIAL SIGNIFICANCE TO THE FIELD

As I stated in my previous review, the work in this manuscript is detailed and quite beautiful. The authors have used several methods to characterize cranial neural crest and ectomesenchymal development in cat shark embryos. The work is unique and provides a foundation (with paired sequencing and expression datasets together with 3D imaging) for other researchers studying evolution of craniofacial diversity, neural crest development, and aquatic species.

SUGGESTIONS TO AUTHORS

The authors have addressed all of my prior concerns. I appreciate their detailed responses to the reviewers and their attempt to improve the manuscript text and figures to better align with the high standards of this journal and other published works and with reviewer concerns. I believe that this manuscript would be a valuable resource for the community.

Reviewer 3

SUMMARY OF THE ADVANCE MADE IN THIS PAPER AND ITS POTENTIAL SIGNIFICANCE TO THE FIELD

This beautiful paper from the Kaucká lab represents an important contribution to the fields of chondrichthyan embryology and vertebrate evo-devo. In my opinion, this resource is deserving of publication in *Development*. In addition to presenting a very nice scRNAseq dataset for a model elasmobranch (among the first of its kind), beautiful plates detailing in situ gene expression for several candidates by HCR and a detailed morphological analysis of ectomesenchymal differentiation and chondrogenesis using SRuCT, the authors have done a commendable job of making these important resources publicly available to the community via a user-friendly web interface. In addition to being an important technical resource, the authors' findings also contribute to an emerging consensus around the developmental basis of morphological diversification within the constraints of deeply conserved developmental GRNs - namely that much of this diversification is attributable to changes in cellular behaviours at relatively "distal" GRN tips.

SUGGESTIONS TO AUTHORS

At this point, the paper has been thoroughly and thoughtfully reviewed by two other experts, and I don't have very much to add beyond what has already been raised and addressed by the authors. I certainly share the first reviewer's enthusiasm for the development of more targeted genetic manipulations in elasmobranch embryos, which would allow for more sophisticated lineage tracing/cell sorting and functional perturbations (indeed, our group is among those working to develop such tools). But I can confirm that Dr. Kaucká's response about current limitations in these systems is accurate: while electroporation is technically feasible, under some limited circumstances, in the embryos of sharks and skates, it is not yet working very well in the early stage embryos that are being investigated by scRNAseq in this paper, nor is it working very efficiently in later developmental stages. Carrying out the sorting/scRNAseq and functional perturbations requested by reviewer 1 would take a significant amount of technical work and troubleshooting (again, many of us are working on these problems, and have been for years...). In my opinion, that is beyond the scope of this paper - and ultimately, I don't think that these additional experiments are required for the specific resource being developed here or for the biological questions that these authors set out to address in the manuscript. From their existing data, the authors can reasonably conclude that core gene expression features of developing neural crest cells are broadly conserved across vertebrates, and that morphological differences between major vertebrate lineages are likely attributable to differences in post-migratory behaviour.

I only have a couple of very minor points to add:

1. In the first line of the abstract (and in the first line of the Discussion), the authors claim that cranial neural crest cells are a jawed vertebrate-specific cell type. This is not quite correct, as cranial neural crest cells are also a conserved feature of cyclostomes. While there has been a stepwise assembly/elaboration of the CNCC GRN through vertebrate evolution, CNCCs are clearly also present in cyclostomes. I recommend that the authors adjust their wording to reflect this.
2. Discussion, Line 421: Delamination of neural crest cells after neural tube closure has also been shown in the little skate (Figure 2 in Gillis et al., 2017, PNAS 114: 13200; and in Figures 1 and 2 in Sleigh and Gillis, 2020, eLife 9: e60635), in case the authors wish to include those citations as well.
3. We have a new paper (Wen et al.) that will be published this week in PNAS showing, among other things, trigeminal, hyoid and branchial NC streams in skate embryos by lineage tracing. If time permits during the production stage, the authors could consider including a citation of this in their discussion around the ancestry of NCC migration in vertebrates.

Thank you for inviting me to review this very nice paper.